# Enterococcus faecalis alters endo-lysosomal trafficking to replicate and persist within mammalian cells

**Ronni A. G. da Silva**[1,2], **Wei Hong Tay**[1], **Foo Kiong Ho**[1], **Frederick Reinhart Tanoto**[1], **Kelvin K. L. Chong**[1], **Pei Yi Choo**[1], **Alexander Ludwig**[3,4], **Kimberly A. Kline**[1,2,3]*

**1** Singapore Centre for Environmental Life Sciences Engineering, Nanyang Technological University, Singapore, **2** Singapore-MIT Alliance for Research and Technology, Antimicrobial Drug Resistance Interdisciplinary Research Group, Singapore, **3** School of Biological Sciences, Nanyang Technological University, Singapore, **4** NTU Institute of Structural Biology, Nanyang Technological University, Singapore

* kkline@ntu.edu.sg

## Abstract

*Enterococcus faecalis* is a frequent opportunistic pathogen of wounds, whose infections are associated with biofilm formation, persistence, and recalcitrance toward treatment. We have previously shown that *E. faecalis* wound infection persists for at least 7 days. Here we report that viable *E. faecalis* are present within both immune and non-immune cells at the wound site up to 5 days after infection, raising the prospect that intracellular persistence contributes to chronic *E. faecalis* infection. Using *in vitro* keratinocyte and macrophage infection models, we show that *E. faecalis* becomes internalized and a subpopulation of bacteria can survive and replicate intracellularly. *E. faecalis* are internalized into keratinocytes primarily via macropinocytosis into single membrane-bound compartments and can persist in late endosomes up to 24 h after infection in the absence of colocalization with the lysosomal protease Cathepsin D or apparent fusion with the lysosome, suggesting that *E. faecalis* blocks endosomal maturation. Indeed, intracellular *E. faecalis* infection results in heterotypic intracellular trafficking with partial or absent labelling of *E. faecalis*-containing compartments with Rab5 and Rab7, small GTPases required for the endosome-lysosome trafficking. In addition, *E. faecalis* infection results in marked reduction of Rab5 and Rab7 protein levels which may also contribute to attenuated Rab incorporation into *E. faecalis*-containing compartments. Finally, we demonstrate that intracellular *E. faecalis* derived from infected keratinocytes are significantly more efficient in reinfecting new keratinocytes. Together, these data suggest that intracellular proliferation of *E. faecalis* may contribute to its persistence in the face of a robust immune response, providing a primed reservoir of bacteria for subsequent reinfection.

## Author summary

*Enterococcus faecalis* is often isolated from chronic wounds. Prior to this study, *E. faecalis* has been observed within different cell types, suggesting that it can successfully colonize

**Funding:** Funding for this work was provided by the National Research Foundation and Ministry of Education Singapore under its Research Centre of Excellence Program, by the National Research Foundation under its Singapore NRF Fellowship program (https://www.nrf.gov.sg/funding-grants/nrf-fellowship) to KAK (NRF-NRFF2011-11), by the Ministry of Education Singapore (https://researchgrant.gov.sg/Pages/GrantCallDetail.aspx?AXID=MOET2EP2-01-2021&CompanyCode=moe) under its Tier 2 programs to KAK (MOE2014-T2-1–129 and MOE2018-T2-1-127), by the Ministry of Education Singapore Tier 1 grants to A.L. (MOE RG136/17 and MOE RG39/14), and by an NTU Start-up grant to AL. RAGDS is supported by the National Research Foundation, Prime Minister's Office, Singapore, under its Campus for Research Excellence and Technological Enterprise (CREATE) program, through core funding of the Singapore-MIT Alliance for Research and Technology (SMART) Antimicrobial Resistance Interdisciplinary Research Group (AMR IRG). The funders had no role in study design, data collection and analysis, decision to publish, or preparation of the manuscript.

**Competing interests:** The authors have declared that no competing interests exist.

intracellular spaces. However, to date, little is known about the mechanisms for *E. faecalis* intracellular survival. Here, we describe key features of the intracellular lifestyle of *E. faecalis*. We show that *E. faecalis* exists in an intracellular state within immune cells and non-immune cells during mammalian wound infection. We show that *E. faecalis* can survive and replicate inside keratinocytes and macrophages, and intracellularly replicating *E. faecalis* are primed to more efficiently cause reinfection, potentially contributing to chronic or persistent infections. To establish this intracellular lifestyle, *E. faecalis* is taken up by keratinocytes primarily via macropinocytosis, whereupon it manipulates the endosomal pathway and expression of trafficking molecules required for endo-lysosomal fusion, enabling *E. faecalis* to avoid lysosomal degradation and consequent death. These results advance our understanding of *E. faecalis* pathogenesis, demonstrating mechanistically how this classic extracellular pathogen can co-opt host cells for intracellular persistence, and highlight the heterogeneity of mechanisms bacteria can use to avoid host-mediated killing.

## Introduction

*Enterococcus faecalis*, traditionally considered an extracellular pathogen, is a member of the healthy human gut microbiome and a frequent opportunistic pathogen of the urinary tract and wounds. Enterococci are one of the most frequently isolated bacterial genera from wound infections [1–5]; however, their pathogenic mechanisms enabling persistence in this niche are not well understood.

We have previously shown in a mouse excisional wound infection model that *E. faecalis* undergo acute replication and long-term persistence, leading to delayed wound healing, despite a robust innate inflammatory response at the wound site [6]. These data suggest that *E. faecalis* possess mechanisms to evade the innate immune response, and indeed, we have also shown that extracellular *E. faecalis* can actively suppress NF-κB activation in macrophages [7]. In addition, *E. faecalis* can persist within a variety of eukaryotic cells including macrophages [8–10], osteoblasts [11,12], monocytes [13], endothelial cells [14], and epithelial cells [15–20]. However, the mechanisms mediating intracellular persistence have not been well studied.

Once internalized, intracellular bacteria can be trafficked via the endo-lysosomal pathway. In this pathway, the small GTPase Rab5 regulates early endosome/macropinosome formation while Rab7, via replacement of Rab5, is required for the maturation of early to late endosomes, as well as for the fusion of late endosomes with lysosomes [21]. Late endosome-lysosome fusion is a critical step for the formation of an acidic and degradative compartment that eliminates bacteria, yet bacteria have evolved multiple mechanisms to interfere with this process [22]. For instance, *Mycobacterium tuberculosis* prevents Rab7 recruitment and, consequently, phagosome maturation, by interfering with Rab5 effectors, which are auxiliary proteins that support Rab5 conversion to Rab7 [23,24]. *Listeria monocytogenes* also inhibits Rab7 recruitment by inhibiting Rab5 GDP exchange activity in host cells [25]. *Coxiella burnetii* can localize to compartments labelled with Rab5 and LAMP1 (a marker of the late endosome/lysosome) but not Rab7 [26,27]. However, very little is known about the intracellular trafficking of *E. faecalis* and whether it can manipulate this pathway for its survival.

In this study, we sought to understand how *E. faecalis* persist within mammalian cells and how intracellularity contributes to pathogenesis. Using a mouse model of wound infection, we found viable *E. faecalis* within both immune and non-immune cells at the wound site up to 5 days after infection and provide evidence that intracellular *E. faecalis* is found in an active state

of replication *in vivo*. Using an *in vitro* model of keratinocyte infection, we show that *E. faecalis* is taken up into these cells via macropinocytosis into single-membrane bound compartments, whereupon they can persist and manipulate the endosomal pathway. We show that internalized *E. faecalis* rarely co-localize with Cathepsin D and a subset of intracellular bacteria ultimately undergoes replication. Interestingly, *E. faecalis* infection results in a marked reduction of Rab5 and Rab7 protein levels, which may explain how *E. faecalis* prevent endo-lysosomal fusion. Finally, we show that *E. faecalis* derived from the intracellular niche are primed to more efficiently reinfect new keratinocytes. Together, our data are consistent with a model in which a subpopulation of *E. faecalis* are taken up into mammalian cells during wound infection, providing immune protection and a replicative niche, which may serve as a nidus for chronically infected wounds.

## Results

### Intracellular *E. faecalis* are present within CD45+ and CD45- cells during mouse wound infection

To determine whether *E. faecalis* persist intracellularly within infected wounds, we infected wounded mice with $10^6$ CFU of *E. faecalis* for 1, 3 and 5 days. Infected wounds were dissociated to a single cell suspension, treated with gentamicin and penicillin G to kill extracellular bacteria, immunolabeled with anti-CD45 antibody, and sorted into CD45+ immune cells and CD45- non-immune cells. These sorted cells were then lysed for the enumeration of intracellular bacteria. Consistent with literature reporting the ability of *E. faecalis* to persist within phagocytic immune cells, we recovered viable *E. faecalis* from CD45+ cells (**Fig 1A**). In addition, intracellular *E. faecalis* was also recovered from the CD45- population, up to 5 days post infection (dpi) (**Fig 1B**). Compared to the approximately $10^5$ CFU total recoverable *E. faecalis* (both extracellular and intracellular) within wounds at 3 and 5 dpi [6], we can estimate that approximately 1–10% of the total recovered bacterial population are intracellular at these time points. These data demonstrate that *E. faecalis* can exist intracellularly during wound infection, implying it is not an exclusive extracellular pathogen.

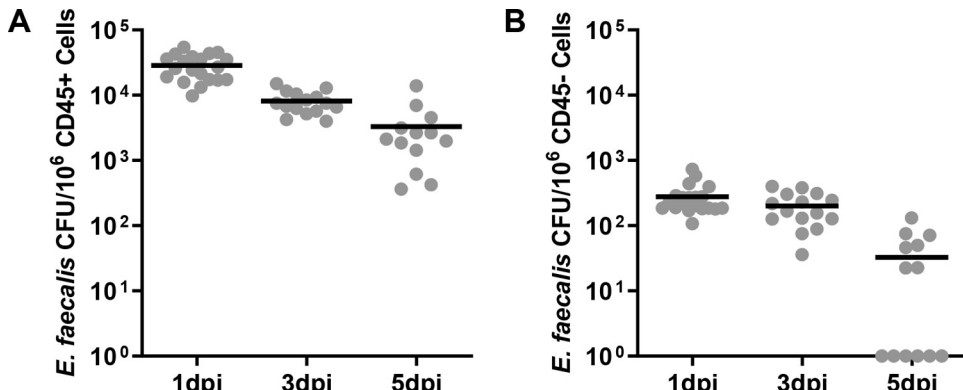

**Fig 1. Recovery of viable *E. faecalis* within host cells during wound infection.** Male C57BL/6 mice were wounded and infected with $10^6$ CFU of *E. faecalis* OG1RF. Wounds were harvested at 1, 3, or 5 days post-infection (dpi), dissociated to single cell suspension, treated with antibiotics to kill extracellular bacteria, labeled, and sorted into **(A)** CD45+ (immune) or **(B)** CD45- (non-immune) populations. CD45+ and CD45- cell populations were then lysed and plated for bacterial CFU. Each data point indicates the CFU within the sorted subpopulation from one mouse. Data shown represent at least 3 independent experiments, each of which included at least 4 mice per time point. Horizontal black lines indicate the mean for each group.

## Actin polymerization and PI3K signaling facilitates uptake of *E. faecalis* into keratinocytes

To investigate the mechanisms by which *E. faecalis* infect non-immune cells at the wound site, we infected the spontaneously immortalized human keratinocyte cell line (HaCaT) with *E. faecalis* strain OG1RF at a multiplicity of infection (MOI) of 1, 10 or 100 for a period of up to 3 hours (h), followed by 1 h of gentamicin and penicillin, and quantified viable intracellular bacteria. The gentamicin and penicillin treatment used was sufficient to kill 99.9% of the extracellular bacteria (**S1A and S1B Fig**). We observed that *E. faecalis* can adhere to keratinocytes at all MOI and time points (**Fig 2A**), and intracellular *E. faecalis* were recovered as early as 1 h post-infection (hpi) at a MOI of 10 and 100 (**Fig 2B**). Parallel cytotoxicity experiments established that *E. faecalis* infection does not negatively affect keratinocytes at early time points of <4 hpi, even in the absence of gentamicin and penicillin (**S2 Fig**). Thus, we chose MOI 100 and no more than 3 h of infection without antibiotics to characterize its intracellular pathogenesis. Since we recovered intracellular *E. faecalis* from infected mouse wounds in both immune and non-immune compartments, we expected to also detect viable *E. faecalis* within mouse fibroblasts and macrophages *in vitro*. Indeed, intracellular *E. faecalis* were recovered from RAW264.7 murine macrophages and NIH/3T3 murine fibroblasts, indicating that *E. faecalis* internalization and persistence is not cell type specific (**S1C Fig**). Similarly, intracellular persistence within keratinocytes was not *E. faecalis* strain specific, as the vancomycin resistant strain V583 persisted at even higher numbers within HaCaT cells after 24 hpi (**S1D and S1E Fig**). *E. faecalis* V583 required 21 h of antibiotic exposure to kill 99.9% of extracellular bacteria, which may extend the effective infection period, but nonetheless supports the conclusion that *E. faecalis* V583 is present intracellularly at 24 hpi (**S1F and S1G Fig**).

Bacterial uptake into non-professional phagocytes such as epithelial cells can proceed via a number of different endogenous endocytic pathways [28]. Previous studies have suggested that *E. faecalis* uptake into non-professional phagocytic cells is dependent on actin and microtubule polymerization, suggestive of macropinocytosis or receptor (clathrin)-mediated endocytosis [14,15]. To determine whether an intact cytoskeleton is important for *E. faecalis* entry into keratinocytes, we pre-treated keratinocytes with specific chemical inhibitors, prior to infection and intracellular CFU enumeration. We found that cytochalasin-D and latrunculin A, inhibitors of actin filament polymerization [29,30], did not alter bacterial adhesion to keratinocytes (**Fig 2C**), but resulted in a significant 100-fold decrease in recoverable intracellular bacteria at 4 hpi, demonstrating that actin polymerization is important for the entry process (**Fig 2D**). By contrast, colchicine, an inhibitor of microtubule polymerization [31], did not impede bacterial adhesion and minimally impacted uptake only at 2 hpi, suggesting that *E. faecalis* may enter keratinocytes via receptor-mediated endocytosis in some cases (**S3A Fig**). Since many endocytic pathways rely on an intact actin cytoskeleton [32], a panel of additional selective inhibitors was used to determine the mechanism of *E. faecalis* entry. Inhibitors of receptor (clathrin)- and caveolae-mediated endocytosis did not meaningfully affect bacterial adhesion or internalization (**S3B and S3C Fig).** Although, we observed a small but statistically significant decrease in the number of internalized bacteria at 3 hpi in dynasore-treated cells, it is likely due to the increased cytotoxicity associated with dynasore treatment (**S1 Table**). To test the role of macropinocytosis in *E. faecalis* uptake, we pre-treated keratinocytes with the phosphoinositide 3-kinase (PI3K) inhibitor wortmannin [33,34]. Wortmannin did not affect *E. faecalis* adhesion to keratinocytes (**Fig 2E**) but resulted in a 10-fold decrease in intracellular CFU, as compared to the untreated controls (**Fig 2F**). All compounds used were non-cytotoxic to mammalian cells (viability ≥80%), except for dynasore which reduced viability to 76% (**S1 Table**). Together, the strong dependence of *E. faecalis* internalization on actin polymerization

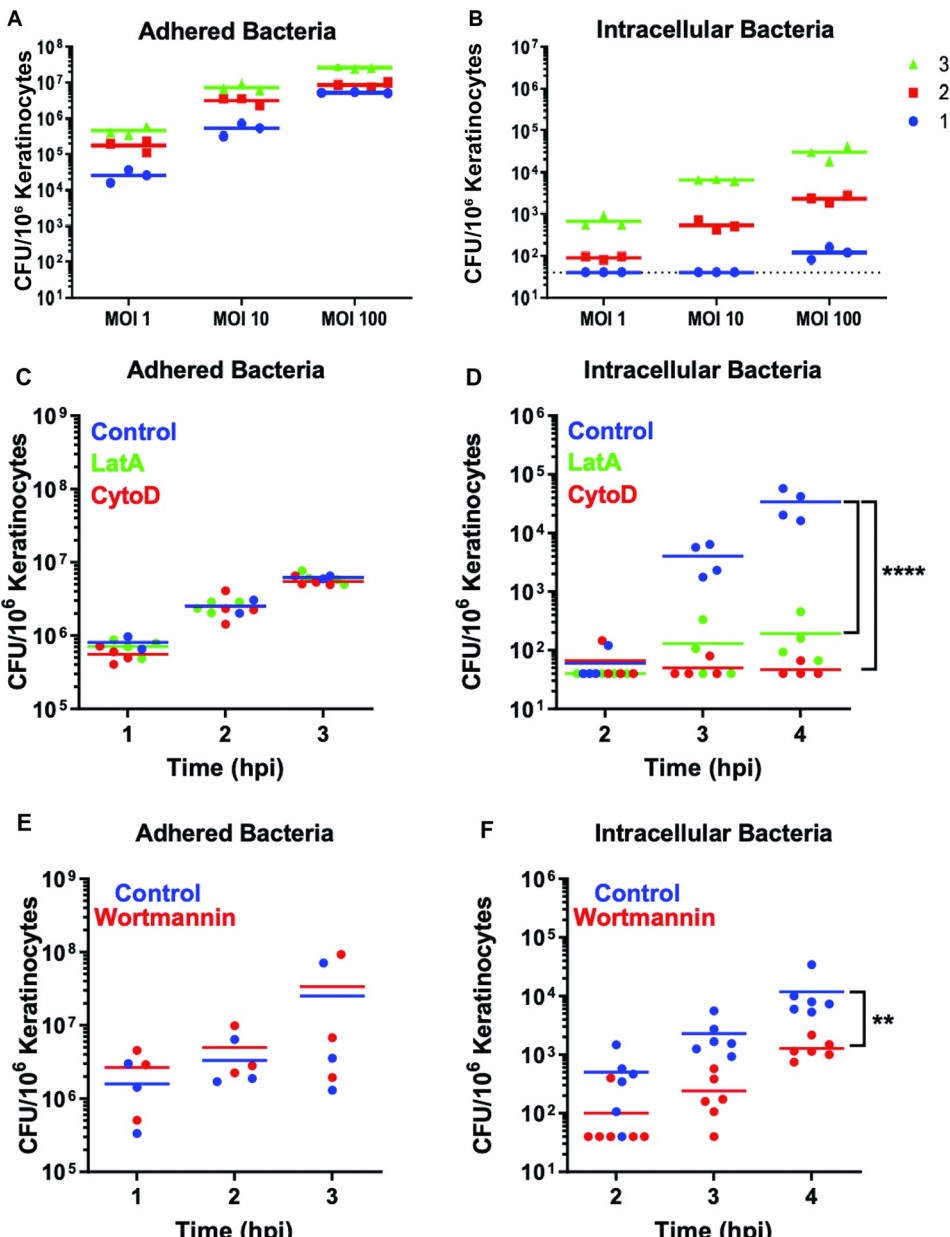

**Fig 2. Time and dose-dependent increase of intracellular *E. faecalis* OG1RF with keratinocytes, following actin polymerization-dependent entry, *in vitro*. (A,B)** $10^6$ keratinocytes were infected with *E. faecalis* OG1RF at the indicated MOI for 1, 2, or 3 h, each followed by another 1 h of antibiotic treatment to eliminate extracellular bacteria. Infected host cells were washed once, lysed, and intracellular CFU enumerated. Solid lines indicate the mean CFU/well from a total of 3 independent experiments. Dashed black line indicates the limit of detection of the assay. **(C-F)** Keratinocytes were pre-treated with actin inhibitors cytochalasin-D (CytoD 1 µg/ml), latrunculin A (LatA 0.25 µg/ml), or PI3K inhibitor wortmannin (0.1 µg/ml) for 0.5 h, followed by *E. faecalis* infection at MOI 100 for 1, 2 or 3 h. Following three PBS washes, cells were lysed and associated adhered bacteria enumerated immediately after infection; or, to quantify intracellular CFU, the initial infection period was followed by 1 h antibiotic treatment, for a total of 2, 3 or 4 hpi, prior to lysis and enumeration. **(C,E)** Adherent or **(D,F)** intracellular bacteria were enumerated at the indicated time points (only significant differences are indicated). Solid lines indicate the mean for each data set of at least 3 independent experiments. **(D)** ****p<0.0001 2 way ANOVA, Tukey's multiple comparisons test. **(F)** **p<0.01 2 way ANOVA, Sidak's multiple comparisons test. (See **S1 Fig** for data related to antibiotic killing efficiency. See **S1 Table** for data related to drug cytotoxicity).

and PI3K and independence of receptor (clathrin)- and caveolae-mediated endocytosis, is consistent with macropinocytosis as a primary means of uptake.

## *E. faecalis* replicates intracellularly *in vitro* and *in vivo*

To confirm the presence of *E. faecalis* within keratinocytes, we imaged keratinocytes infected with *E. faecalis* expressing chromosomally encoded green fluorescent protein (GFP) [35] by confocal laser scanning microscopy (CLSM). Images taken at 4 hpi, from cells infected for 3 h followed by 1 h gentamicin and penicillin treatment to kill extracellular bacteria, revealed 1–10 intracellular bacteria within each infected keratinocyte (**Fig 3A**). This observation suggested either that selected infected keratinocytes can take up many *E. faecalis*, or that *E. faecalis* could replicate within these keratinocytes. We extended the period of post-infection antibiotic exposure up to 24 hpi and recovered similar intracellular CFU within the whole population (**S1D Fig**). However, within single infected keratinocytes, we visualized 10–30 *E. faecalis*, which often clustered in a perinuclear region (**Fig 3B**). At the same 24 hpi time point, we also detected clusters of fluorescent *E. faecalis* peripheral to apparently apoptotic keratinocytes (**S4 Fig**), indicative of intracellular bacteria that have either escaped from the keratinocyte or of bacteria derived from lysed keratinocytes, of which the latter may account for the slight decrease in overall intracellular CFU over time.

To directly examine intracellular replication of *E. faecalis* OG1RF, we treated infected HaCaT and RAW264.7 cell lines with BrdU, a nucleotide analogue that is incorporated into replicating DNA; and RADA, a TAMRA-based fluorescent D-amino acid that labels newly synthesized peptidoglycan and has been recently used to assess *E. faecalis* replication within hepatocytes [36–38]. As a control, *E. faecalis* treated with a bacteriostatic concentration of the antibiotic ramoplanin to halt replication do not incorporate fluorescent D-amino acids [39] or BrdU (**S5A and S5B Fig**). Following 3 h infection and 1 h antibiotic treatment, HaCaT and RAW264.7 cells were treated with BrdU or RADA for another 20 h concomitantly with antibiotics, to ensure that only intracellular replicating bacteria could incorporate the compounds. CLSM images of intracellular bacteria in both cell lines confirmed that *E. faecalis* incorporated both compounds, indicating *E. faecalis* were in a state of active replication (**Fig 3C**). We observed 45% (16/35) of the infected HaCaT contained *E. faecalis* that had incorporated RADA. To determine whether *E. faecalis* can replicate intracellularly *in vivo*, we infected mouse excisional wounds with *E. faecalis* expressing episomally encoded GFP (pDasherGFP) for 24 h. BrdU was injected and applied topically to the infected wound 1.5 h prior to wound collection. CLSM analysis of *ex vivo* wounds that were dissociated to multicellular clusters and immunolabeled for CD45 expression showed clusters of BrdU positive intracellular bacteria within CD45- cells (**Figs 3D and S6 and S1 Video**). Altogether these observations suggest that *E. faecalis* can replicate intracellularly within mammalian cells of both immune and epithelial origin.

## Intracellular *E. faecalis* display heterotypic trafficking through early and late endosomes

Since we observed large numbers of *E. faecalis* within keratinocytes in the perinuclear region at 24 hpi, we hypothesized that *E. faecalis* may be trafficked through the host endo-lysosomal pathway. To interrogate this hypothesis, we infected cells for 30 min, 1 hpi and 3 hpi and subsequently immunolabelled the early endosome Rab5 GTPase in infected cells to visualize early intracellular endo-lysosomal compartments. Fluorescence histograms were generated for individual *E. faecalis*-containing compartments and were further validated by visualization of orthogonal views and 3D projections. At 30 min, 1 hpi, and 3 hpi we observed that 31% (13/

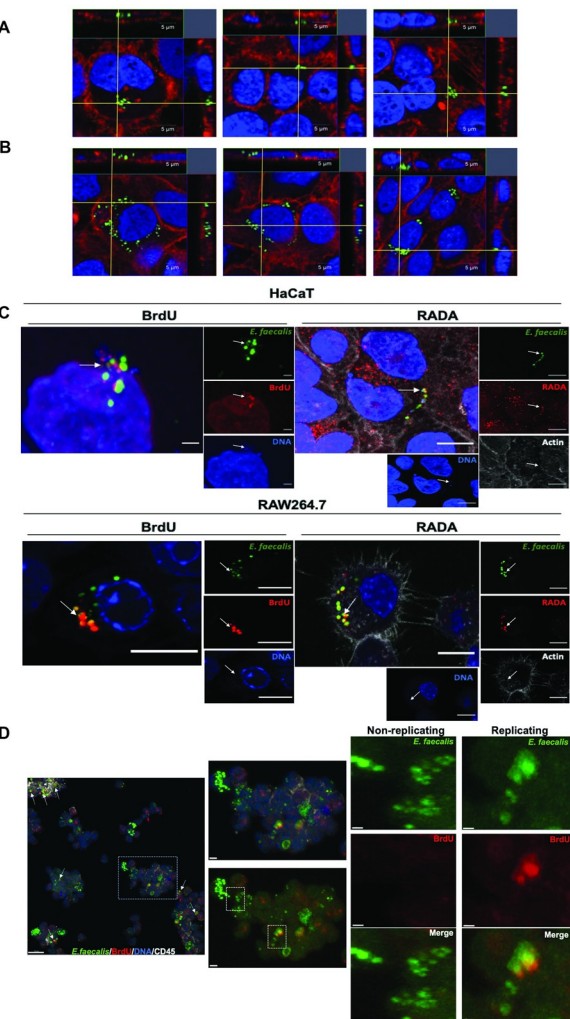

**Fig 3. Intracellular *E. faecalis* replicates *in vitro* in HaCaT and RAW264.7 cells and *in vivo* in CD45-negative cells.**
(**A**) CLSM orthogonal view of internalized *E. faecalis* within HaCaT keratinocytes at 4 hpi (3 h infection + 1 h antibiotic treatment). (**B**) CLSM orthogonal view of internalized *E. faecalis* within HaCaT keratinocytes at 24 hpi (3 h infection + 21 h antibiotic treatment). (**A, B**) Blue, dsDNA stained with Hoechst 33342; green, *E. faecalis*; red, F-actin. Images are representative of 3 independent experiments. Scale bar: 5 μm (**C**) CLSM view of internalized *E. faecalis* stained with BrdU and RADA. Examples of replicating *E. faecalis* are indicated with a white arrow. Blue, dsDNA stained with Hoechst 33342; green, E-GFP, *E. faecalis*; red, BrdU or RADA; white, F-actin. Images are representative of at least 3 independent experiments. Scale bar: 10 μm. (**D**) CLSM view of *ex vivo* murine wound tissue cells following infection and BrdU treatment. Left panel shows examples of potentially replicating *E. faecalis* clusters, indicated with white arrows. Scale bar: 10 μm. Right panels show magnified areas of interest. The marked areas with dashed lines white square show CD45-negative *E. faecalis* containing cells. Scale bars: 2 μm and 0.5 μm. Blue, dsDNA stained with Hoechst 33342; green, *E. faecalis*; red, BrdU; white, CD45. Images are representative of 3 independent experiments. (**S1 Video** shows a 3D video projection of this image. Additional images of *ex vivo* cell experiments are provided in **S6 Fig**).

42), 28% (18/64), and 35% (35/111) of intracellular *E. faecalis*, respectively, were found in Rab5 + labelled compartments (**Figs 4A and S6**). These data suggest that *E. faecalis*-containing compartments that are not in association with Rab5 may either traffic quickly through Rab5+ compartments or avoid association with Rab5 entirely. Additionally, observations of single enterococcal chains, for which we expected uniform association with Rab5, revealed instances of non-uniform Rab5 association along the chains, suggesting either incomplete Rab5

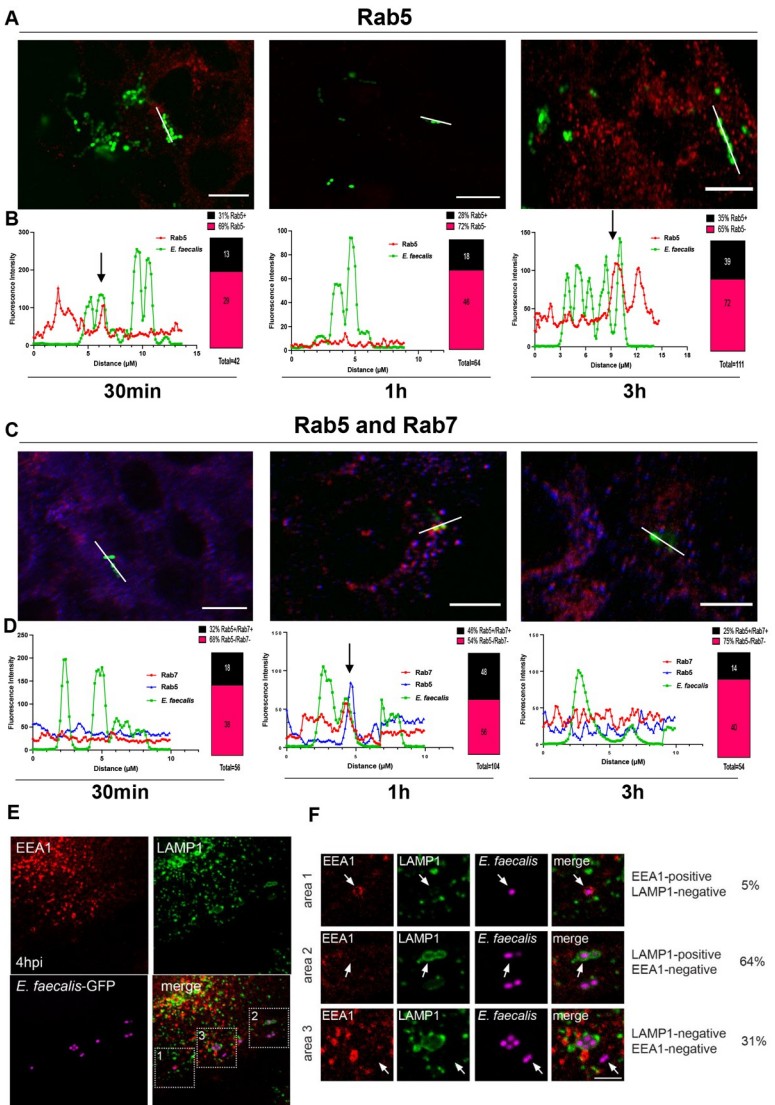

**Fig 4. Intracellular *E. faecalis* displays heterotypic trafficking through early and late endosomes in keratinocytes.**
**(A)** CLSM of infected keratinocytes labelled with antibodies against Rab5 (Alexa fluor 568, red, early endosome) at 30 min, 1 hpi and 3 hpi. **(B)** Representative fluorescence intensity profiles of immunolabeled Rab5 and fluorescent *E. faecalis* (pDasherGFP, green) was assessed over a linear segment of *E. faecalis* cells (derived from the white lines shown in panel A). Labeling of individual cells was manually scored using a combination of histogram overlap (where each cell is a peak on the histogram) and visualization of orthogonal views and 3D projections generated using Imaris 9.0.2. Arrows indicate points of colocalization. Images are representative of three independent experiments. Percentage of *E. faecalis* associated with Rab5 was derived from a minimum of 8 individual confocal images per time point (2–3 images per biological replicate). **(C)** CLSM of infected keratinocytes labelled with antibodies against Rab5 (Alexa Fluor 568, blue; early endosome) and Rab7 (Alexa Fluor 647, red; late endosome) at 30 min, 1 hpi and 3 hpi. **(D)** Representative fluorescence intensity profiles of immunolabeled Rab5 immunolabeled Rab7 and *E. faecalis* (pDasherGFP, green) was assessed over a linear segment (derived from the white lines shown in panel C) and scored as described for panel B. Arrow indicates points of colocalization. Images are representative of three independent experiments. Percentages are derived from a minimum of 8 individual confocal images per time point. CLSM of infected HaCaTs immunolabeled for EEA1 (early endosome) and LAMP1 (late endosome/lysosome) at 4 hpi. **(F)** Magnified images of boxed areas in (A) showing representative images of *E. faecalis* containing compartments. Percentages are derived from 10 individual confocal images and a total of 55 *E. faecalis* diplococci. Images are maximum intensity projections of 4–5 optical sections (~2 μm z-volume) and are representative of 3 independent experiments. Scale bars: A: 10 μm; C: 10 μm; E: 10 μm; F: 2 μm. (See **S7 and S8 Figs** for additional representative images).

immunolabelling, or different Rab5 interaction or immunolabelling efficiency at different parts of the chain (**Fig 4B**). To investigate this further, we immunolabelled Rab5 at 30 min, 1 hpi and 3 hpi concomitantly with Rab7, a late endosomal GTPase that ultimately replaces Rab5 as the endo-lysosomal pathway progresses from early to late endosomes [40]. We observed 32% (18/56), 46% (48/104) and 25% (14/54) of intracellular *E. faecalis* in compartments labelled for both Rab5 and Rab7 at 30 min, 1 hpi, and 3 hpi, respectively (**Figs 4C and 4D, and** S7). Looking at Rab7 alone (which may include instances of Rab5 proximal labelling), we observed that 67% (44/65), 54% (63/116) and 63% (40/63) of *E. faecalis*-containing compartments lacked Rab7 labelling at 30 min, 1 hpi, and 3 hpi, respectively. These data support our hypothesis that some intracellular *E. faecalis* may be escaping Rab5/7 compartments altogether. In addition, at 4 hpi or 24 hpi we also immunolabeled the infected cells to visualize the following intracellular endo-lysosomal compartments: EEA1 (early endosome antigen 1, early endosome), Rab7 and LAMP1 (lysosomal-associated membrane protein, late endosome/pre-lysosome) (**Figs 4E, 4F and** S8). By 4 hpi, although EEA1-labeled early endosomes were often observed in close proximity to *E. faecalis*-containing compartments, only 5% (3/55) of internalized bacteria compartments showed a clear association with EEA1, and these compartments did not contain LAMP1 (EEA1+/LAMP1-). Instead, 64% (35/55) of internalized *E. faecalis* were in compartments associated with LAMP1 but not EEA1 (EEA1-/LAMP1+). The remainder (31% of internalized bacteria (17/55)) was neither associated with LAMP1- nor EEA1-labeled compartments (EEA1-/LAMP1-) (**Fig 4E and 4F**). Furthermore, we observed that Rab7 was associated with 28% (16/58) of *E. faecalis*-containing compartments at 4 hpi and with 32% (10/31) of *E. faecalis*-containing compartments at 24 hpi (S8A **Fig**); however, some but not all LAMP1+ *E. faecalis*-containing compartments were also Rab7+ (S8B **Fig**), again suggesting a degree of heterogeneity among the intracellular niche of *E. faecalis*. Altogether, these data suggest that a subset of internalized *E. faecalis* traffic rapidly through early endosomes and reach late endosomal compartments as early as 30 min post-infection and LAMP1+ late endosomal compartments by 4 hpi. Another pool of *E. faecalis* may avoid the canonical Rab5/Rab7 endo-lysosomal pathway entirely.

## Intracellular *E. faecalis* escapes lysosomal fusion with late endosome compartments

While *E. faecalis* can survive in murine macrophages by resisting acidification, which in turn prevents fusion with lysosomes [10], this has not been previously documented in epithelial cells. Our results showing that *E. faecalis* can replicate in keratinocytes led us to hypothesize that fusion between late endosomes and lysosomes may be impeded, permitting intracellular survival. To determine if lysosomes fuse with *E. faecalis*–containing late endosomes, we first immunolabeled infected cells at 24 hpi for LAMP1 (with a polyclonal antibody) and the lysosomal protease Cathepsin D. We found that LAMP1 and Cathepsin D colocalized in infected and non-infected cells, as expected. However, when looking at *E. faecalis*-containing compartments, while 47% (30/64) of internalized *E. faecalis* were observed in LAMP1+ compartments, these compartments were conspicuously devoid of Cathepsin D (**Fig 5A, white arrows**). The remainder (53% of internalized bacteria (34/64)) did not colocalise with either LAMP1 or Cathepsin D. We validated this finding using a monoclonal antibody against LAMP1 concomitantly with Cathepsin D immunolabeling. We confirmed that the majority of intracellular *E. faecalis* escaped lysosomal fusion with only 4% (3/69) and 8% (4/50) of the observed intracellular compartments containing *E. faecalis* labelled with Cathepsin D and LAMP1 simultaneously at 4 hpi and 24 hpi, respectively (**Figs 5B, 5C, and** S9). Finally, we also observed a complete lack of colocalization between *E. faecalis*-containing compartments and M6PR (mannose-

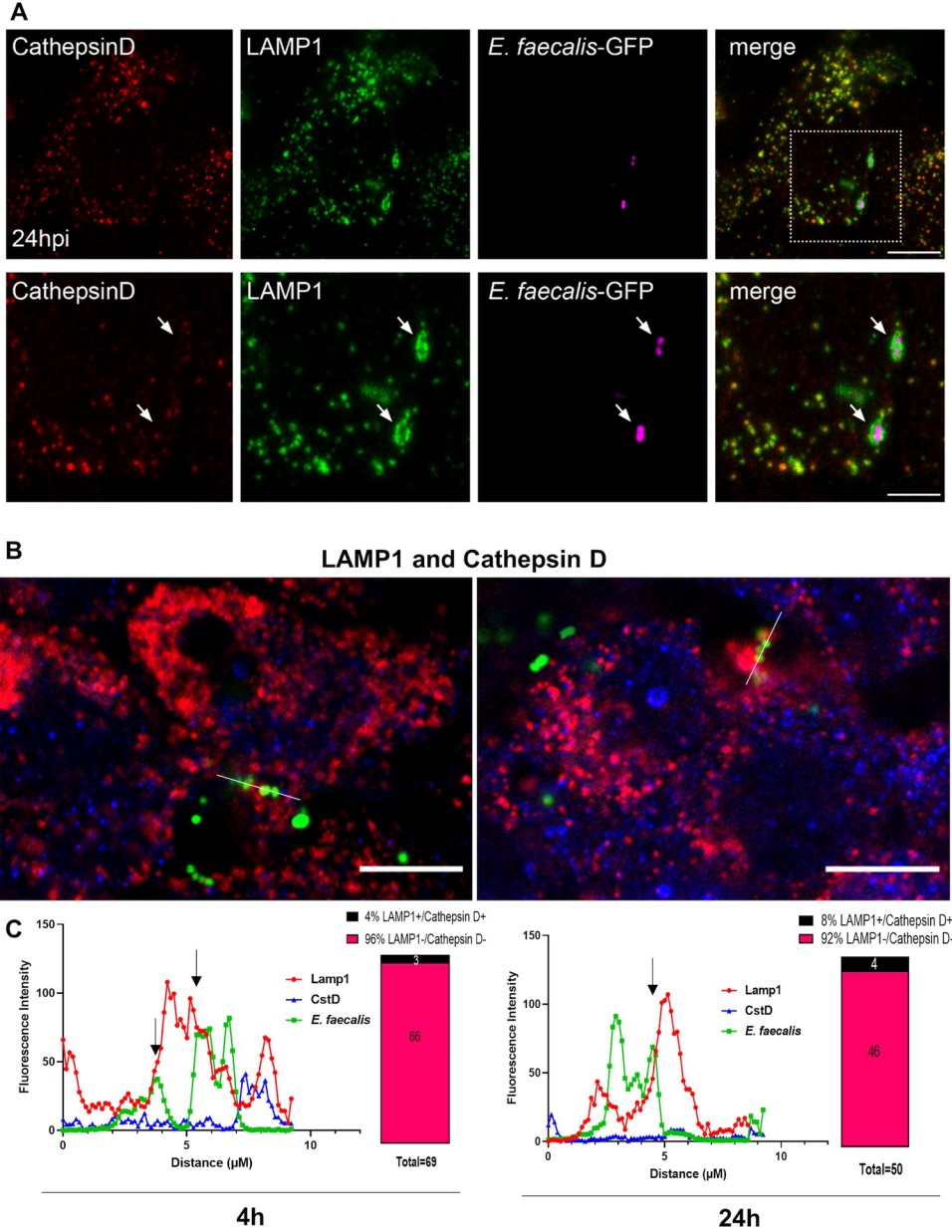

**Fig 5. Intracellular *E. faecalis* escape lysosomal fusion with late endosome compartments. (A)** CLSM of infected keratinocytes stained for the lysosomal protease Cathepsin D (late endosome/lysosome) and LAMP1 (late endosome/lysosome; polyclonal antibody) at 24 hpi. Bottom panel shows the boxed region above. Images are maximum intensity projections of 4–5 optical sections (~2 μm z-volume) and are representative of 3 independent experiments. Scale bars: top panel: 10 μm; C bottom panel: 5 μm. **(B)** CLSM of infected keratinocytes stained for the lysosomal protease Cathepsin D (late endosome/lysosome) and LAMP1 (late endosome/lysosome; monoclonal antibody) at 4 hpi and 24 hpi **(C)** Fluorescence intensity of Cathepsin D (Alexa Fluor 568, blue), LAMP1 (Alexa Fluor 647, red) and *E. faecalis* (pDasherGFP, green) were assessed over a linear segment (histograms) and scored as labelled by visualization of orthogonal views and 3D projections on Imaris 9.0.2. Arrow indicates points of colocalization. Images are representative of three independent experiments. Measurements (percentages) are derived from a minimum of 8 individual confocal images per time point (2–3 images per biological replicate).

6-phosphate receptor, a late endosome/pre-lysosome marker), which delivers lysosomal hydrolases to pre-lysosomal compartments (**S10 Fig**). Importantly, LAMP1+ compartments containing *E. faecalis* often appeared distended, particularly at 24 hpi (**Figs 5A and S10**).

Based on these observations, we conclude that *E. faecalis* escapes lysosomal fusion. Moreover, we propose that intracellular replication occurs within late endosomes until a bacterial threshold is reached, whereupon the compartment is unable to accommodate additional bacteria leading to compartment and/or cell lysis.

## *E. faecalis* intracellular infection reduces Rab5 and Rab7 protein levels

Rab5 and Rab7 are small GTPases that are critical for the formation of early and late endocytic compartments. To test whether *E. faecalis* infection affects Rab protein levels, we analyzed Rab5 and Rab7 proteins in the infected keratinocyte population, following infection with either *E. faecalis* strain OG1RF or strain V583. At 4 hpi, *E. faecalis* infection with either strain resulted in significantly lower Rab5 protein levels (**Fig 6A and 6B**). Somewhat unexpectedly, although we observed Rab7 associated with some *E. faecalis* compartments, we also observed a global reduction in Rab7 protein levels for both strains at 4 hpi. While Rab7 levels were restored by 24 hpi for both strains, Rab5 in V583-infected keratinocytes remained lower at 24

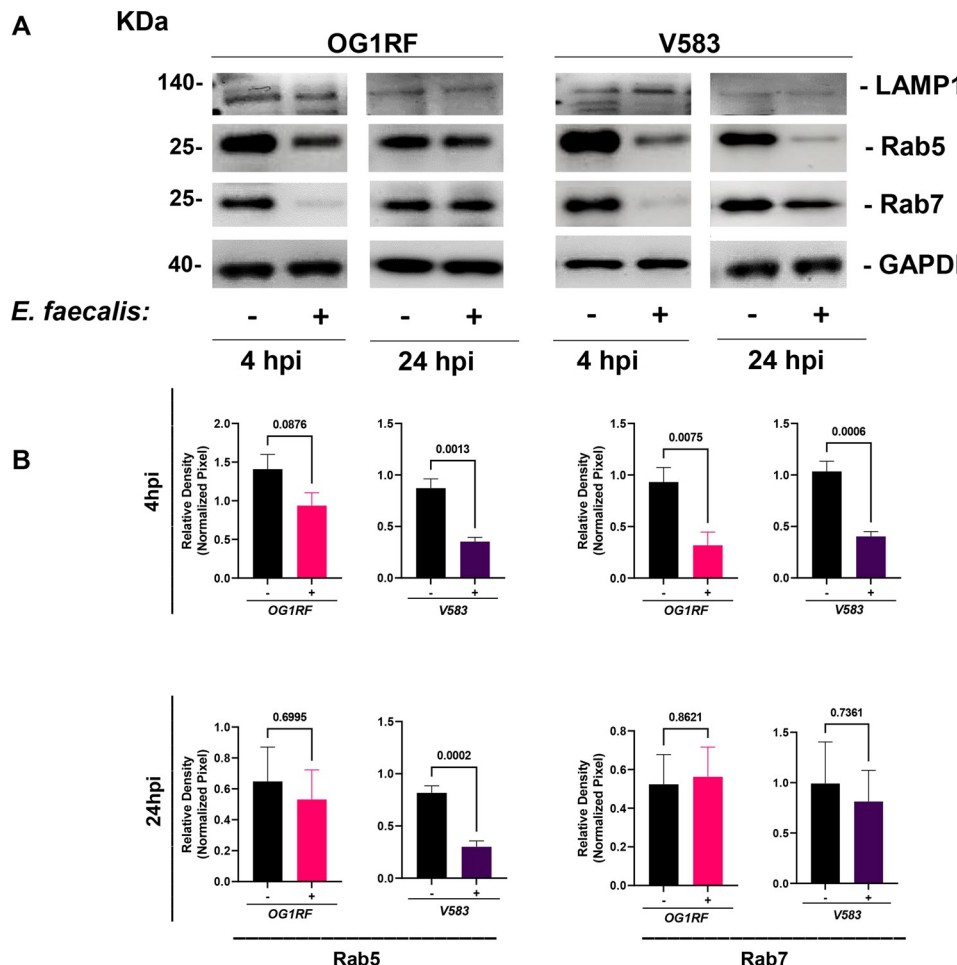

**Fig 6.** *E. faecalis* **infection reduces keratinocyte expression of both Rab5 and Rab7 at 4 hpi. (A)** Whole cell lysate analyzed by immunoblotting with antibodies α-LAMP1 (polyclonal antibody), α-Rab5, α-Rab7 and α-GAPDH. HaCaT cells were incubated with (+) and without (-) *E. faecalis* OG1RF and V583 for 4 hpi and 24 hpi. Images shown are representative of at least 3 biological replicates. **(B)** Relative density of the bands of interest were normalized against loading control (GAPDH). Error bars represent biological replicates and mean ± SEM from at least 4 independent experiments. Statistical analysis was performed using unpaired T-test with Welch's correction.

hpi, which may correlate with greater CFU and intracellular survival for V583 compared to OG1RF (**S1F and S1G Fig**). The expression of other endo-lysosomal proteins, such as Cathepsin D and LAMP1, were unchanged upon *E. faecalis* infection (**S11 Fig**), indicating that *E. faecalis* selectively interferes with the levels of endosomal Rab5 and Rab7 proteins. Taken together, the combined *E. faecalis*-mediated reduction in Rab expression, coupled with the ability of nearly 70% of *E. faecalis*-containing compartments to avoid Rab7 recruitment (**Figs 4 and** S8) could explain the lack of colocalization of Cathepsin D with *E. faecalis*-containing compartments (**Fig 5**) and is consistent with the conclusion that most *E. faecalis*-containing compartments do not fuse with lysosomes. Infected and non-infected HaCaT cells were both maintained under the same antibiotic treatment for protein collection, therefore observed differences in protein levels are not a consequence of antibiotic exposure.

## *E. faecalis* survives in heterogeneously labelled intracellular niches

To visualize the association between *E. faecalis*-containing compartments and endo-lysosomal organelles with greater resolution, we turned to correlative light and electron microscopy (CLEM). We created a HaCaT cell line that stably expresses LAMP1-mCherry and infected these cells with GFP-expressing *E. faecalis* for 18 h (3 h followed by 15 h antibiotic treatment). Confocal microscopy of fixed cells enabled us to locate *E. faecalis* and LAMP1+ compartments in infected cells before processing them for serial section transmission electron microscopy (TEM) (**Figs 7**, **S12, and S13**). These experiments revealed several features of *E. faecalis* intracellular infection that we could not appreciate using fluorescence microscopy alone. First, we observed *E. faecalis* in LAMP1+ compartments (8/26 or 30% of the observed intracellular *E. faecalis*) as well as in vacuoles that appeared to be devoid of LAMP1 (18/26) (**Fig 7A–7C**), which is in line with our immunofluorescence microscopy data (**Figs 4E and 4F, 5, and** S8B). Second, and importantly, regardless of the degree of colocalization with LAMP1, *E. faecalis*-containing vacuoles were invariably bounded by a single membrane (**Fig 7D–7H**). In addition, most internalized bacteria appeared to be morphologically intact, with a uniform density and a clearly defined septum and bacterial envelope (**Figs 7D–7H and S13A**). Third, we did not find examples of multiple replicating *E. faecalis* within a single LAMP1+ compartment leading to membrane distension, as predicted by our immunofluorescence imaging (**Figs 4E and 4F, 5 and S10**). Rather, we observed at most two diplococci within a single compartment (**Figs 7D and S13B**). Finally, although, we observed that LAMP1+ electron dense compartments of unknown nature (potentially lysosomes which are LAMP1+ organelles with dense ultrastructural appearance) were often located in close proximity to *E. faecalis* containing vacuoles, we did not observe any obvious fusion events between the two compartments (**Fig 7C and 7E**). In some instances, however, vacuoles harbouring *E. faecalis* appeared to contain LAMP1 multi-lamellar bodies (MLBs) (**Figs 7G, S12 and S13**). Together, these EM data confirm both that internalized *E. faecalis* can survive in late endosomal organelles and that there is heterogeneity within the intracellular niches for this organism. Furthermore, *E. faecalis*-containing vacuoles do not exhibit lysosomal features and do not appear to fuse with lysosomes. These findings raise the possibility that *E. faecalis* could be hijacking the endo-lysosomal pathway, altering organelle identity to prevent lysosomal recognition, and allowing for intracellular survival, replication and eventual escape.

## Intracellular *E. faecalis* is primed for more efficient reinfection

To investigate whether internalization of *E. faecalis* into keratinocytes provides an advantage for subsequent reinfection, we harvested intracellular bacteria and measured its ability to reinfect keratinocytes. An initial infection was performed at MOI 50 for 3 h to isolate intracellular

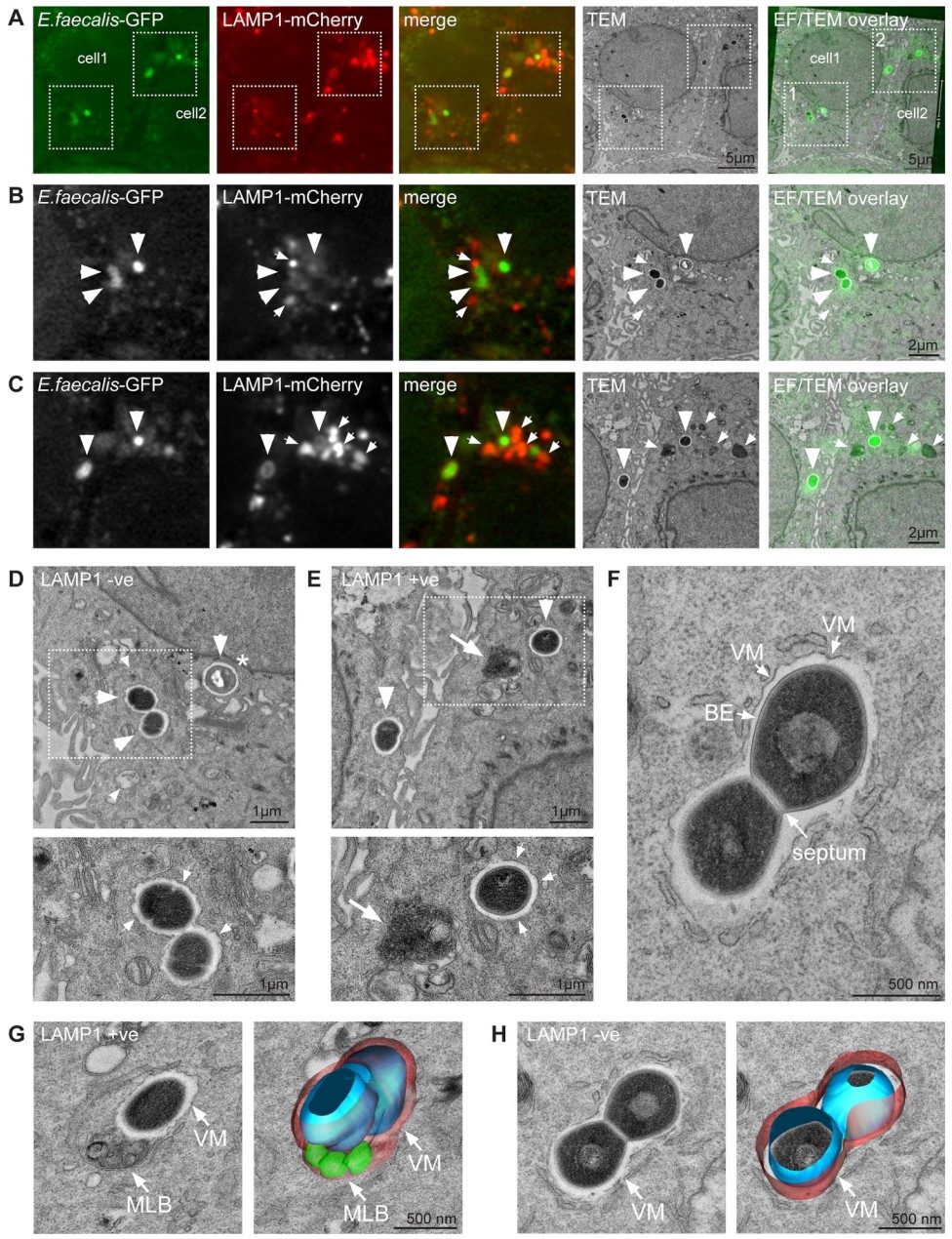

**Fig 7. Correlative light and electron microscopy of *E. faecalis* infected keratinocytes. (A)** Spinning disk confocal microscopy and correlative TEM of HaCaTs stably expressing LAMP1-mCherry infected with *E. faecalis*-GFP at 18 hpi. Confocal images are maximum intensity projections of 4–5 optical sections (~2 μm z-volume). **(B and C)** Enlarged views of the two areas highlighted in (A); panels B and C show the boxed areas 1 and 2 in (A), respectively. Large arrowheads indicate *E. faecalis* containing vacuoles, small arrows indicate LAMP1+ (LAMP1+ve) compartments. Note that *E. faecalis* is present in LAMP1- (LAMP1-ve) vacuoles in (B), and in LAMP1+ (LAMP1+ve) vacuoles in (C). LAMP1+ve compartments appear electron-dense (see TEM panel in C). * indicates a bacterium with altered appearance, possibly due to partial degradation. **(D and E)** Representative high magnification TEM images of LAMP1-ve (D) and LAMP1+ve *E. faecalis* containing vacuoles corresponding to data shown in (B) and (C), respectively. The large arrow in (E) indicates an electron-dense LAMP1+ve compartment in close proximity to an *E. faecalis* containing vacuole. Arrowheads in the lower panels indicate the presence of a single layer membrane surrounding the bacterial vacuole. **(F)** High magnification view of an *E. faecalis* containing vacuole. The vacuolar membrane (VM), the bacterial envelope (BE), and the septum are indicated. **(F and G)** 3D surface rendering of representative *E. faecalis* containing vacuoles reconstructed from serial TEM sections. An *E. faecalis* containing vacuole containing a LAMP1+ve multilamellar body (MLB) is shown in (G), while the vacuole shown in (F) is LAMP1-ve and does not contain a MLB. (See **S12 Fig** for data related to F-H, and **S13 Fig** for data related to D).

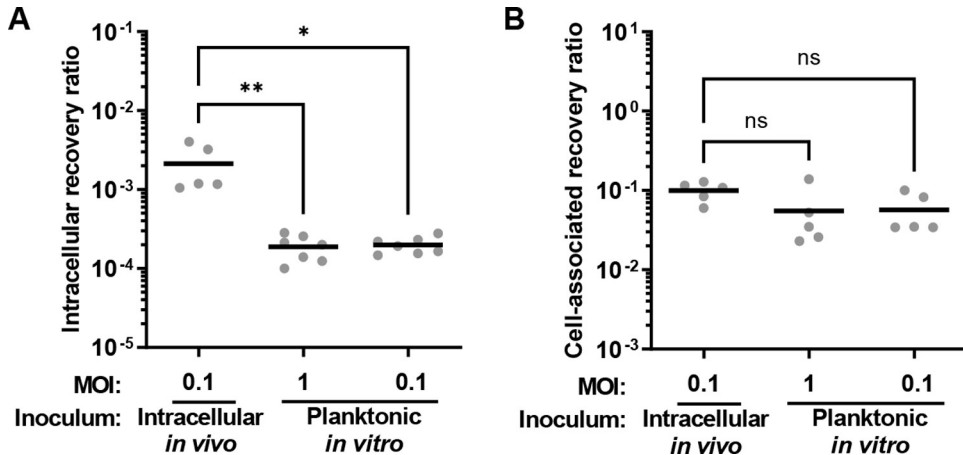

**Fig 8. Increased recovery of internalized *E. faecalis* upon reinfection of keratinocytes by intracellular bacteria.**
Intracellular bacteria were isolated from infected keratinocytes and used as the inoculum (intracellular *in vivo*) for reinfection of new monolayers of keratinocytes. Parallel infections at various MOI were performed using *E. faecalis* not yet exposed to keratinocytes, grown planktonically *in vitro* as the inoculum. **(A)** Infections proceeded for 3 h followed by 1 h of antibiotic exposure to kill the extracellular bacteria prior to intracellular CFU enumeration. **(B)** Infections proceeded for 3 h prior extensive washing to remove non-adhered extracellular bacteria, prior to enumeration of total cell-associated (both extracellular adhered and intracellular) CFU. Each circle represents CFU data averaged from 3 separate wells from a single biological experiment, showing a total of 5–7 independent experiments. Data are represented in the figure as the recovery ratio, which for **(A)** intracellular is the CFU recovered divided by the inoculum CFU and for **(B)** cell-associated is the CFU recovered divided by the non-cell-associated extracellular CFU in the same well, to account for cell growth during the assay. Horizontal black line indicates the mean for each condition. *p<0.05, **p<0.01 Kruskal Wallis test with Dunn's post test.

bacteria. Intracellular-derived bacteria were then used for reinfection of keratinocytes at an MOI of 0.1, the highest MOI practically attainable given the low intracellular CFU, for another 3 h. 1 h of gentamicin and penicillin treatment was performed after both the initial infection and the second round of infection. Internalization recovery ratios were determined by comparing inoculum CFU to intracellular CFU bacteria during the reinfection assay. Parallel experiments with *E. faecalis* not yet exposed to keratinocytes at comparable MOI showed that reinfection with intracellular-derived bacteria resulted in significantly higher internalization rates, as shown by the recovery ratio (**Fig 8A**). Intracellular growth exclusively promoted reinfection, because total cell associated bacteria comprising both adherent and intracellular bacteria, was not significantly different from a planktonically grown inoculum (**Fig 8B**). These results are similar to observations made in *S. pyogenes*, where longer periods of internalization in macrophages increased recovered CFU during subsequent reinfections [41]. Taken together, these data suggest that internalized *E. faecalis* can more efficiently reinfect host cells.

## Discussion

*E. faecalis* is among the commonly isolated microbial species cultured from chronic wound infections. The ability of *E. faecalis* to persist in the face of a robust immune response and antibiotic therapy is frequently attributed to its ability to form biofilms during these infections. However, a number of bacterial pathogens undertake an intracellular pathway during infection that can contribute to persistent and or recurrent infection. This is well-described for uropathogenic *E. coli* (UPEC), particularly in animal models in which UPEC can replicate to high numbers within urothelial cells as intracellular bacterial communities or can persist in a quiescent intracellular state within LAMP1+ compartments for long periods of time, promoting recurrent and chronic infection [42–45]. While there are numerous reports of intracellular *E.*

*faecalis* within a variety of non-immune cells [13–20], the contribution of an intracellular life-cycle to *E. faecalis* infection has been minimally investigated. Here, we report that, *in vitro*, *E. faecalis* become internalized into keratinocytes primarily via macropinocytosis, whereupon they undergo heterotypic trafficking through the endosomal pathway, which enables their replication and survival. These findings raise the possibility that this intracellular lifecycle may be linked to persistent and chronic infections, such as those that occur in wounds. Further, we demonstrate that intracellularity may be physiologically relevant in a mouse model of wound infection, where *E. faecalis* exists within both immune and non-immune cells for at least 5 days after infection. Importantly, *E. faecalis* recovered from within keratinocytes are primed to more efficiently infect new keratinocytes to seed another round of infection.

Previous studies using either professional or non-professional phagocytic cell lines have reported the internalization, but not the replication of intracellular *E. faecalis* [9,13,15] as these studies used only antibiotic protection assays coupled with TEM at single time points. Here, we performed antibiotic protection assays coupled with imaging across multiple time points, and our results similarly show that *E. faecalis* can enter and survive intracellularly up to 72 hpi. Importantly, we show with BrdU labelling that *E. faecalis* can be found in a state of active replication in cells harvested from infected wounds. This finding is further supported by *in vitro* analyses of intracellular *E. faecalis* from infected keratinocytes and macrophages that were incubated with BrdU and RADA. This is the first reported evidence, to our knowledge, of *E. faecalis* intracellular replication within epithelial cells or macrophages. Consistent with our observation, *E. faecalis* has also been shown to replicate within human hepatocytes *in vitro* and has been observed as clusters in association with hepatocytes in a mouse model of intravenous infection [38]. Together these data suggest that once *E. faecalis* enters mammalian cells, at least some of the bacteria are able to replicate intracellularly. Other "classical" extracellular bacteria including *S. aureus* and *P. aeruginosa* are also able to replicate intracellularly [46–51]. Similar studies have also shown that *S. pyogenes* can be taken up by both immune and non-immune cells, where it can replicate, survive host defenses and disseminate to distant sites [52,53].

In this work, we show that *E. faecalis* enters keratinocytes in a process that is dependent on actin polymerization and PI3K signalling, and independent of receptor (clathrin)- or caveolae-mediated endocytosis. Chemical inhibition of actin polymerization by cytochalasin D and PI3K signaling by wortmannin specifically affects macropinocytosis but not receptor (clathrin)-mediated endocytosis [54–56]. These findings suggest that *E. faecalis* strain OG1RF enters keratinocytes primarily in a macropinocytotic process. A previous study suggested that clinical isolates of *E. faecalis* enter HeLa (human epithelioid carcinoma) cells via either macropinocytosis or clathrin-mediated endocytosis, supported by inhibitors of microtubule polymerization and cytosolic acidification that reduced intracellular CFU [15]. We also observe that *E. faecalis* uptake is transiently diminished when receptor (clathrin)-mediated endocytosis is inhibited, suggesting that *E. faecalis* may also take advantage of this uptake pathway in some instances. Thus, it may be that different strains of *E. faecalis* favor entry into mammalian cells by different mechanisms, and *E. faecalis* OG1RF used in this study preferentially enters via macropinocytosis. However, another study reported that *E. faecalis* OG1 strain derivatives, closely related to OG1RF, entered human umbilical vein endothelial cells (HUVEC) cells via receptor (clathrin)-mediated endocytosis, in a cytocholasin D- and colchicine-dependent manner [14]. Because Millan *et al* used similar drug concentrations as we did, we suggest that OG1-related strains may enter epithelial cells primarily via macropinocytosis and endothelial cells primarily via receptor (clathrin)-mediated endocytosis.

Additionally, once inside keratinocytes, at least some *E. faecalis* commence trafficking through the endosomal pathway. As soon as 30 min after infection, most *E. faecalis*-containing compartments lacked the early endosome marker Rab5. Moreover, we did not observe any

labelling with Rab7 in nearly 70% of the intracellular compartments containing internalized enterococci at time points <3 hpi. At 4 hpi, the majority (60–70%) of internalized *E. faecalis* were in compartments that were heterogeneously positive for the late endosomal markers LAMP1 and/or Rab7. These data indicate that Rab5 labelling may be incomplete or that *E. faecalis* associate with Rab5-containing compartments quickly and transiently, or only in a subset of infected cells, in either case with no subsequent or delayed Rab7 labelling. At the same time, since we observed some Rab5+/Rab7+ compartments at all early time points, traditional Rab5/Rab7 conversion dynamics from early to late endosome also happens in a subset of infected cells. Taken together, these data point to the possibility that at least a subset of internalized *E. faecalis* enter Rab5+/EEA1+ early endosomes/macropinosomes, and rapidly transit into late endosomal compartments. In parallel, Rab5 and Rab7 protein levels in infected keratinocytes were markedly decreased in comparison to non-infected keratinocytes. We predict that *E. faecalis* infection-driven reduction in Rab expression is crucial to determine the outcome of *E. faecalis* intracellular survival since Rab5 and Rab7 control important fusion events between early and late endosomes and late endosomes and lysosomes [22]. Rab GTPases are commonly hijacked by bacteria to promote their survival [57]. For comparison, intracellular microbes such as *M. tuberculosis* and *L. monocytogenes* distinctively modify the Rab5 machinery arresting phagosome maturation [22]. *C. burnetii* prevents Rab7 recruitment [58] and *B. cenocepacia* affects Rab7 activation [59]. However, to the best of our knowledge, our data are the first to show differences in overall Rab5 and Rab7 protein levels as a potential bacterial subversion mechanism for the macropinosome. Studies are underway to determine the bacterial factors and mechanisms by which *E. faecalis* affects Rab protein levels. While *E. faecalis*-containing LAMP1+ compartments appeared distended at 24 hpi, many *E. faecalis* were not tightly associated with LAMP1 or Rab7. Furthermore, we rarely observed Cathepsin D in *E. faecalis*-containing compartment, suggesting that late endosomes containing internalized bacteria could be missing markers or that these late endosomal compartments have been modified, making lysosomal fusion a rare event. Other intracellular pathogens such as *C. burnetii* and *Francisella tularensis* also reside in compartments devoid of Cathepsin D, or in compartments with very low levels of Cathepsin D [60,61]. The authors suggest that this was achieved by escaping fusion with lysosomes. In support of this view, TEM revealed that all membrane-bound *E. faecalis* were spatially separated from LAMP1+ electron dense compartments of unknown nature (potentially a lysosome, which is a LAMP1+ organelle), and there was no indication of membrane fusion between *E. faecalis*-containing compartments and lysosomes. Notably, we observed some *E. faecalis* in association with LAMP1 and Rab7, suggesting that some internalized *E. faecalis* cells may transit via the normal endocytic pathway and fuse with lysosomes. Collectively our results support a model in which late endosomes containing *E. faecalis* are modified, preventing the expected destruction of intracellular *E. faecalis* by lysosomal fusion and allowing them to replicate from within.

We propose three potential fates for different subsets of internalized *E. faecalis* (Fig 9). 1) Macropinosome maturation into Rab7+/LAMP1+ late endosomes and fusion with the lysosome, leading to the degradation of *E. faecalis* (**Fig 9B-I**). 2) Macropinosome maturation into LAMP1+ but Rab7- compartments, leading to *E. faecalis* survival because Rab7 presence is required for lysosome fusion (**Fig 9B-II**). 3) Macropinosome maturation into compartments lacking both Rab7 and LAMP1, which would also lead to *E. faecalis* survival (**Fig 9B-III**). Additionally, we cannot exclude the possibility that *E. faecalis*-containing compartments initially contain both Rab7 and LAMP1 late endosomal markers, but are subsequently modified by *E. faecalis* to increase its survival. In other words, there could be a transition from **I** into **II** and/or **III** mediated by downregulation of Rab5 and Rab7. Furthermore, intracellular *E. faecalis* that do not associate with any examined endo-lysosomal markers could reflect a cytosolic

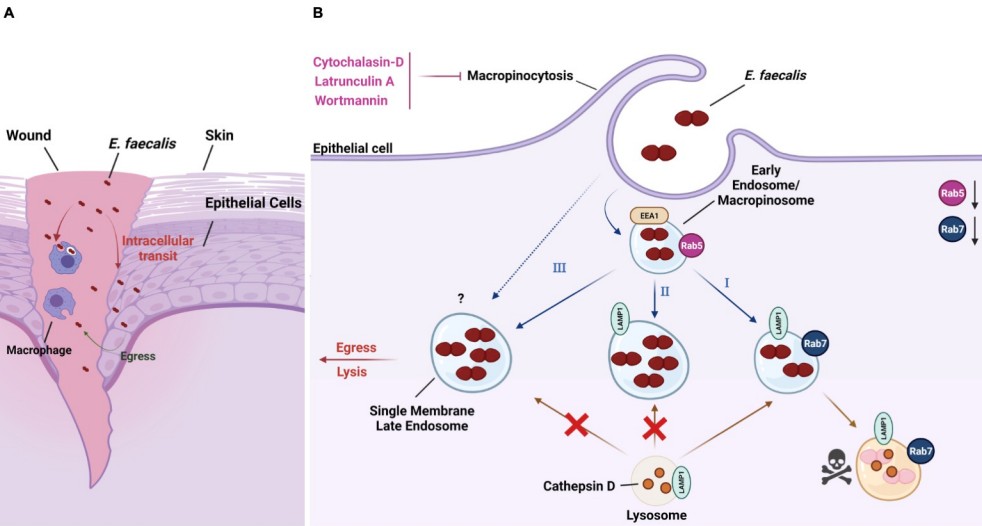

**Fig 9. The intracellular lifestyle of *E. faecalis*. (A)** *E. faecalis* can survive intracellularly either inside keratinocytes or macrophages and contribute to reinfection in the wound site. **(B)** *E. faecalis* is taken up by keratinocytes via macropinocytosis. Treatment of keratinocytes with macropinocytosis inhibitors such as cytochalasin D, latrunculin A and wortmannin prevents *E. faecalis* uptake. **(I)** *E. faecalis* was observed inside single membrane endosomes that were positive to different endosomal proteins indicating that *E. faecalis* may transit through the normal endocytic pathway inside keratinocytes. **(II and III)** *E. faecalis* interferes with Rab5 and Rab7 proteins levels which could help prevent the late endosome to fuse with lysosome. The low percentage of compartments positive to Rab5 and then Rab7 combined with the varied percentages of LAMP1+ compartments indicates that *E. faecalis* may be affecting the expected macropinosome transit contributing to *E. faecalis* intracellular survival. Created with BioRender.com.

state. Finally, we have also shown that infected host cells can eventually die, releasing *E. faecalis* into the periphery of dead host cells. Based on reinfection studies, we propose that those bacteria released from dead cells may be primed to infect other cells, resulting in an enhanced cycle of reinfection. Altogether, our work has demonstrated that *E. faecalis* can enter, survive, replicate and escape from keratinocytes *in vitro*. If this intracellular lifecycle also exists *in vivo*, and extends to other cell types such as macrophages as our data suggest, these findings may allow for an abundant protective niche for bacterial persistence that could contribute to the chronic persistent infections associated with *E. faecalis*.

## Materials and methods

### Ethics statement

All procedures, including isoflurane for anesthesia and $CO_2$ followed by cervical dislocation for euthanasia, were approved and performed in accordance with the Institutional Animal Care and Use Committee (IACUC) in Nanyang Technological University, School of Biological Sciences (ARF SBS/NIEA-0314).

**Bacterial strains and growth conditions.** *Enterococcus faecalis* strain OG1RF [62] and its derivative strains, including OG1RF strain SD234 which chromosomally expresses GFP [35] and the plasmid-based fluorescent *E. faecalis* OG1RF (pDasherGFP) (built using the IP-Free Fluorescent ProteinPaintbox -*E. coli*) [63], and *Enterococcus faecalis strain* V583 [64] were grown using Brain Heart Infusion (BHI) broth and agar (Becton, Dickinson and Company, Franklin Lakes, NJ). Unless otherwise stated, bacterial strains were streaked from glycerol stocks stored at -80˚C, inoculated and grown overnight statically for 16–20 h in 20 ml of liquid BHI broth. Cells were harvested by centrifugation at 5000 RPM (4˚C) for 4 min. The supernatant was discarded, and the pellet washed with 1 ml of sterile phosphate buffered saline (PBS).

The pellet was then resuspended in sterile PBS to an optical density ($OD_{600nm}$) of 0.7 for *E. faecalis*, equivalent to $2–3×10^8$ colony forming units (CFU).

**Mouse wound excisional model.** The procedure for mouse wound infections was modified from a previous study [6]. Briefly, male wild-type C57BL/6 mice (7–8 weeks old, 22 to 25 g; InVivos, Singapore) were anesthetized with 3% isoflurane. Following dorsal hair trimming, the skin was then disinfected with 70% ethanol before creating a 6-mm full-thickness wound using a biopsy punch (Integra Miltex, New York, USA). *E. faecalis* corresponding to $2–3 \times 10^6$ CFU was added to the wound site and sealed with a transparent dressing (Tegaderm 3M, St Paul Minnesota, USA). At the indicated time points, mice were euthanized and a 1 cm by 1 cm squared piece of skin surrounding the wound site was excised and collected in sterile PBS. Skin samples were homogenized, and the viable bacteria enumerated by plating onto both BHI plates and rifampicin (50 μg/ml) selection plates to ensure all recovered CFU correspond to the inoculated strain.

**Fluorescence-activated cell sorting (FACS).** Excised skin samples were harvested, placed in 1.5 ml Eppendorf tubes containing 2.5 U/ml liberase prepared in DMEM (10% FBS) with 500 μg/ml of gentamicin and penicillin G (Sigma-Aldrich, St. Louis, MO) and minced with surgical scissors. The mixture was then transferred into 6-well plates and incubated for 1 h at 37˚C in a 5% $CO_2$ humidified atmosphere with constant agitation. Dissociated cells were then passed through a 70 μm cell strainer to remove undigested tissues and spun down at 1350 RPM for 5 min at 4˚C. The enzymatic solution was then aspirated, and cells were blocked in 500 μl of FACS buffer (2% FBS (Gibco, Thermo Fisher Scientific, Singapore), 0.2 mM ethylenediaminetetraacetic acid (EDTA) in PBS (Gibco, Thermo Fisher Scientific, Singapore). Cells were then incubated with 10 μl of Fc-blocker (anti-CD16/CD32 antibody) for 20 min, followed by incubation with an anti-mouse CD45-Cy7 conjugated antibody (BD Pharmingen, Singapore) (1:400 dilution) for 20 min at room temperature. Cells were then centrifuged at 1350 RPM for 5 min at 4˚C and washed in FACS buffer before a final resuspension in FACS buffer. Following which, cells were sorted using a BD FACSAria 3 sorter, equipped with 4 air-cooled lasers (355 nm UV, 488 nm Blue, 561 nm Yellow/Green and 633 nm Red) (Becton Dickinson, Franklin Lakes, NJ). Post-sorting, cells were lysed with 0.1% Triton X–100 (Sigma-Aldrich, St. Louis, MO) and intracellular bacteria plated onto both BHI plates and antibiotic selection plates to ensure all recovered CFU correspond to the inoculated strain. Supernatants were also plated to ensure that no bacteria were present post-sorting.

**BrdU Labelling.** An intraperitoneal injection of an aqueous solution of BrdU (120 mg/kg) at 90 min before death, in addition to 10 μL of a 10 μM solution of BrdU spotted on the wound site was performed as adapted from Mysorekar and Hultgren [42]. After sacrifice, excised wounds were placed in DMEM (Gibco) medium containing 500 μg/ml of gentamicin and penicillin G, andtreated with 0.5 mg/ml Liberase TL (Roche) for 1 h to create a single cell suspension. Cells were then fixed in 4% paraformaldehyde (PFA), solubilized in 2 N HCl solution at 37˚C for 30 min, washed three times with PBS (Gibco) and further stained with rat α-CD45 (Abcam, Cambridge, UK) antibody for 1 h followed by three times PBS washes and addition of secondary antibody α-rat-Alexa Fluor 568 (Abcam, Cambridge, UK) for 1h. Secondary antibody was then washed off with PBS three times and samples were incubated with a rat α-BrdU-Alexa Fluor 647 (Abcam, Cambridge, UK) conjugate was added overnight in 0.1% Triton X–100 as per α-BrdU antibody manufacturer instructions. After the α-BrdU antibody was washed off with PBS three times, samples were incubated with a 1:1000 dilution of Hoechst 33342 for 5 min at room temperature with subsequent PBS washing three times. Cells were mounted with ProLong Glass Antifade Mountant (Invitrogen, Thermo Fisher Scientific, Singapore).

**Cell culture.** The spontaneously immortalized human keratinocyte cell line, HaCaT (AddexBio, San Diego, CA) and RAW264.7 murine macrophage-like cell line (InvivoGen, Singapore) was cultured at 37˚C in a 5% $CO_2$ humidified atmosphere. All cells were grown and maintained in Dulbecco's modified Eagle's medium (DMEM) (Gibco; Thermo Fisher Scientific, Singapore) with 10% heat-inactivated fetal bovine serum (FBS) (PAA, GE Healthcare, Singapore), and 100 U of penicillin–streptomycin (Gibco, Thermo Fisher Scientific, Singapore) where appropriate for extracellular bacterial killing. The culture medium was replaced once every three days, and upon reaching 80% confluency, cultures were passaged. Passaging was achieved by treatment with 0.25% trypsin-EDTA (Gibco; Thermo Fisher Scientific, Singapore) for 6 min and seeding cells at a density of $2\times10^6$ cells/T75 flask (Nunc; Thermo Fisher Scientific, Singapore). For RAW264.7 cells, passaging was achieved by gentle cell scraping and seeding cells at a density of $3\times10^6$ cells/T75 flask (Nunc; Thermo Fisher Scientific, Singapore).

**Intracellular infection assay.** HaCaT keratinocytes were seeded at a density of $5\times10^5$ cells/well in a 6-well tissue culture plate (Nunc; Thermo Fisher Scientific, Singapore) and grown for 3 days at 37˚C in a 5% $CO_2$ humidified atmosphere. After 3 days, each well had approximately $1–1.5\times10^6$ keratinocytes. Keratinocytes were infected at a multiplicity of infection (MOI) of 100, 10 or 1 for up to 3 h. Following infection, the media was aspirated, and the cells were washed three times in PBS and either lysed in 0.1% Triton X–100 (Sigma-Aldrich, St. Louis, MO) for enumeration of cell-associated/adhered bacteria, or incubated with 500 μg/ml of gentamicin and penicillin G (Sigma-Aldrich, St. Louis, MO) in complete DMEM for 1–70 h to selectively kill extracellular bacteria. The antibiotic containing medium was then removed and the cells were washed 3 times in PBS before the intracellular bacteria was enumerated. For macrophages, RAW264.7 cells were seeded at a density of $1\times10^6$ cells/well in a 6-well tissue culture plate (Nunc; Thermo Fisher Scientific, Singapore) or $3\times10^5$ cells/well in a 24-well tissue culture plate (Nunc; Thermo Fisher Scientific, Singapore) and allowed to attach overnight at 37˚C in a 5% $CO_2$ humidified atmosphere. Infection was performed similarly as described above.

**Chemical inhibition of endocytosis.** All chemical inhibitors were purchased from Sigma-Aldrich (St. Louis, MO), unless otherwise stated. Stock solutions of cytochalasin D (1 mg/ml), latrunculin A (100 μg/ml), colchicine (10 mg/ml), dynasore (25 mg/ml), nystatin (25 mg/ml) and wortmannin (10 mg/ml) were dissolved in DMSO unless otherwise indicated and stored at −20˚C. Pharmacological inhibitors were added to cells 30 min prior to any infection and maintained throughout the course of the infection. Actin polymerization was inhibited by 1 μg/ml of cytochalasin D [65,66] or 250 ng/ml of latrunculin A [29,67–69]. Microtubule polymerization was inhibited by 10 μg/ml of colchicine [70–72]. PI3K was inhibited by 0.1 μg/ml of wortmannin [66,70,73,74]. The large GTPase dynamin that is important for the formation of clathrin-coated vesicles was inhibited by 25 μg/ml of dynasore [75–78]. The caveolae-mediated endocytosis was disrupted by 25 μg/ml of nystatin [75,79–81]. Addition of pharmacological inhibitors at the concentrations indicated had no effect on bacteria viability, where growth kinetics and CFU count were similar to the untreated bacteria control. All chemical inhibitors were purchased from Sigma-Aldrich (St. Louis, MO), unless otherwise stated.

**Immunofluorescence staining.** HaCaT or RAW264.7 cells were seeded at $3\times10^5$ cells/well into a 24-well culture plate with 10 mm coverslips, and allowed to attach overnight at 37˚C, 5% $CO_2$ in a humidified incubator. Infection with *E. faecalis* was performed as described previously. Following infection, coverslips seeded with cells were washed 3 times in PBS and fixed with 4% PFA at 4˚C for 15 min. Cells were then permeabilized with 0.1% Triton X–100 (Sigma-Aldrich, St. Louis, MO) (actin) or 0.1% saponin (endosomal compartments) for 15 min at room temperature and washed 3 times in PBS or PBS with 0.1% saponin, respectively. Cells were then blocked with PBS supplemented with 0.1% saponin and 2% Bovine Serum

Albumin (BSA). For actin labelling, the phalloidin–Alexa Fluor 568 conjugate (Thermo Fisher Scientific, Singapore) was diluted 1:40 in PBS. For antibody labelling of endosomal compartments, antibody solutions were diluted in PBS with 0.1% saponin at a 1:10 dilution for mouse α-LAMP1 (ab25630, Abcam, Cambridge, UK), 1:50 mouse mAb α-LAMP1 (D4O1S, Cell Signalling Technology, USA), 1:50 for rabbit α-EEA1-Alexa Fluor 647 (ab196186, Abcam, Cambridge, UK), 1:100 for rabbit α-M6PR-Alexa Fluor 568 (ab202535, Abcam, Cambridge, UK), 1:30 for rabbit α-Rab5 (ab218624, Abcam, Cambridge, UK), 1:30 for rabbit α-Rab7 (ab137029, Abcam, Cambridge, UK), 1:30 for rabbit α-Rab7-Alexa Fluor 647 (ab198337, Abcam, Cambridge, UK) or a 1:100 for rabbit α-cathepsin D (ab75852, Abcam, Cambridge, UK) and incubated overnight at 4˚C. The following day, coverslips were washed 3 times in 1× PBS with 0.1% saponin and incubated with a 1:500 dilution of the following secondary antibodies (Thermo Fisher Scientific, Singapore): goat α-Mouse IgG (H+L) Alexa Fluor Plus 647, goat α-Rabbit IgG (H+L) Alexa Fluor Plus 647, goat anti-Rabbit IgG (H+L) Alexa Fluor 568, goat anti-Mouse IgG (H+L) Alexa Fluor 568 for 1 h at room temperature. Coverslips were then washed 3 times in 1× PBS with 0.1% saponin and incubated with a 1:500 dilution of Hoechst 33342 (Thermo Fisher Scientific, Singapore) for 20 min at room temperature. Next, the coverslips were subjected to a final wash, 3 times with PBS with 0.1% saponin and 2 times with PBS. After washing, the coverslips were mounted with SlowFade Diamond Antifade (Thermo Fisher Scientific, Singapore) and sealed. In instances in which the host for the antibodies coincided (e.g. rabbit α-Rab5 and rabbit α-Rab7-Alexa Fluor 647), primary and secondary antibodies were added for one labelling first, washed off with PBS and then a conjugated antibody was used for the second labelling of interest to avoid overlap with the previous secondary antibody. LAMP1 was visualized using polyclonal mouse α-LAMP1 (ab25630, Abcam, Cambridge, UK) or monoclonal mouse mAb α-LAMP1 (D4O1S, Cell Signalling Technology, USA), as indicated in the figure legends.

**BrdU Labelling of infected HaCaT and RAW264.7 cells.** Intracellular infection for 3 h with fluorescent *E. faecalis* OG1RF (pDasherGFP) at MOI 100 for HaCaT cells and MOI 1 for RAW264.7 cells was followed by 1 h antibiotic treatment to kill extracellular bacteria. Complete DMEM was replaced with complete DMEM containing a 10 μM BrdU solution (ab142567, Abcam, Cambridge, UK) and 500 μg/ml of gentamicin and penicillin G for another 20 h. Cells were then fixed in 4% PFA, solubilized in 2 N hydrochloric acid (HCL) solution at 37˚C for 30 min, washed three times with PBS (Gibco) and further stained with rat α-BrdU-Alexa Fluor 647 (ab220075, Abcam, Cambridge, UK) conjugate overnight in 0.1% Triton X–100 as per manufacturer instructions. After α-BrdU antibody was washed with PBS three times, samples were incubated with a 1:1000 dilution of Hoechst 33342 for 5 min at room temperature. Finally, samples were subjected to a final wash, three times with PBS. Cells were mounted with ProLong Glass Antifade Mountant (Invitrogen, Thermo Fisher Scientific, Singapore).

**Peptidoglycan labelling (RADA labelling) of intracellular bacteria.** Infection was carried out as described for BrdU labeling. Complete DMEM was replaced with complete DMEM containing 250 μM orange-red TAMRA-based fluorescent D-amino acid (RADA, Tocris) solution and 500ug/ml of gentamicin and penicillin G for another 20 h. The RADA solution was removed, cells were washed with PBS three times and then fixed in 4% PFA. For actin labelling, phalloidin–Alexa Fluor 647 conjugate (Thermo Fisher Scientific, Singapore) was diluted 1:40 in 0.1% Triton X–100 solution and incubated for a minimum of 1 h. The phalloidin solution was washed off with PBS three times and samples were then incubated with a 1:1000 dilution of Hoechst 33342 for 5 min at room temperature. Finally, samples were subjected to a final wash, three times with PBS. Cells were mounted with ProLong Glass Antifade Mountant (Invitrogen, Thermo Fisher Scientific, Singapore). To test if non-replicating *E. faecalis* is incorporating BrdU or RADA, following the protocol used in [39], *E. faecalis* was

treated with the bacteriostatic antibiotic ramoplanin to halt replication. Briefly, exponentially growing cells of *E. faecalis* were harvested then allowed to grow in the presence or absence of ramoplanin (26 μg/ml) with BrdU or RADA for 1 h. BrdU and RADA labelled samples were then processed as per their labelling protocol.

**Confocal Laser Scanning Microscopy (CLSM).** Confocal images were acquired on a 63× oil objective (NA 1.4, Plan Apochromat, Zeiss) fitted onto an Elyra PS.1 with LSM 780 confocal unit (Carl Zeiss, Göttingen, Germany) using the Zeiss Zen Black 2012 FP2 software suite. Laser power and gain were kept constant between experiments. Labelling experiments for control primary and secondary antibodies alone were performed in parallel infected cells. Z-stacked images were processed using Zen 2.1 (Carl Zeiss, Göttingen, Germany). Acquired images were visually analyzed using Imaris x64 9.0.2 (Oxford Instruments). Representative fluorescence intensity profiles of immunolabeled proteins were also assessed over a linear segment (histograms) using Fiji [82]. Individual cells were manually scored as labelled by a combination of histogram overlap (where each cell is a peak on the histogram) and visualization of orthogonal views and 3D projections on Imaris x64 9.0.2.

**Construction of LAMP1-mCherry strain.** pLAMP1-mCherry (Addgene plasmid #45147, Addgene, Cambridge) and EF1α-mCherry-N1 plasmid (Thermo Fisher Scientific, Singapore) were isolated using the Monarch Plasmid Miniprep Kit (New England BioLabs Inc., USA), according to manufacturer's instructions. The LAMP1 gene was then sub cloned into the pEF1α-mCherry-N1 vector using the In-Fusion HD Cloning Kit (Clontech, Takara, Japan), according to manufacturer's instructions. The plasmid construct was then transformed into Stellar competent cells by incubating at 42°C for 1 min. Transformed colonies with the desired construct was assessed by colony PCR. Primers used for the cloning and subsequent verification are shown in **S2 Table**. EF1α Lamp1-mCherry plasmid was extracted from successful transformants with the Monarch Plasmid Miniprep Kit (New England BioLabs Inc., USA), according to manufacturer's instructions. Keratinocytes were grown in 6-well tissue culture plates as described above, where each well was seeded with $2\times10^5$ cells. 2.5 μg of plasmid DNA was transfected into keratinocytes using Lipofectamine 3000 (Invitrogen; Thermo Fisher Scientific, Singapore), according to manufacturer's instructions. The culture media was replaced after 6 h of incubation, followed by a subsequent replacement 18 h later. Keratinocytes were then subjected to Geneticin selection (1 mg/ml) (Invitrogen; Thermo Fisher Scientific, Singapore) to select for transfected clones. Clones stably overexpressing Lamp1-mCherry were subjected to validation by immunoblotting and flow cytometry. Clonal populations were selected and subjected to fluorescence activated cell sorting (FACS) to ensure that the entire population were expressing the fluorescent construct.

**Correlative light and electron microscopy.** *E. faecalis* infected keratinocytes stably transfected with LAMP1-mCherry were grown in 35 mm glass bottom dishes (MatTec Corp., Ashland, USA). Cells were fixed at 18 hpi for 2 h on ice in 2.5% glutaraldehyde (EMS) in 0.1 M cacodylate buffer (CB; EMS) pH 7.4 supplemented with 2 mM CaCl$_2$. Cells were washed several times in CB and imaged using a spinning disk confocal microscope (CorrSight, Thermo Fisher Scientific, Singapore). Confocal z-stacks were acquired with a 63x oil objective (NA 1.4, Plan Apochromat M27, Zeiss) on an Orca R2 CCD camera (Hamamatsu, Japan) using standard filter sets. Cells were then further processed for TEM. Briefly, cells were post-fixed with 1% osmium tetroxide for 1 h on ice, washed several times, and incubated with 1% low molecular weight tannic acid ($(C_{14}H_{10}O_9)_n$; EMS) for 1 h at RT. Cells were dehydrated using a graded ethanol series (20%, 50%, 70%, 90%, 100%), and embedded in Durcupan resin (Sigma Aldrich). Areas of interest were sawed out of the dish and sectioned parallel to the glass surface by ultramicrotomy (EM UC7, Leica) using a diamond knife (Diatome). Serial 70–80 nm thin sections were collected on formvar- and carbon-coated copper slot grids (EMS). Electron

micrographs were recorded on a Tecnai T12 (Thermo Fisher Scientific) TEM operated at 120 kV using a 4k x 4k Eagle (Thermo Fisher Scientific) CCD camera. TEM and confocal microscopy images were manually overlaid and aligned in Photoshop with minimal warping or stretching. Serial sections were aligned manually in Photoshop, followed by surface rendering in IMOD.

**Immunoblotting.** Whole cell (WC) lysates were prepared by adding 488 µl of RIPA buffer (50 mM Tris-HCl, pH 8.0; 1% Triton X–100; 0.5% Sodium deoxycholate; 0.1% SDS; 150 mM NaCl) to the wells after intracellular infection assays, where cells were scraped and kept in RIPA buffer for 30 min at 4˚C. Prior to the addition of 74.5 µl of 1 M DTT and 187.5 µl NuPAGE LDS Sample Buffer (4X) (Thermo Fisher Scientific, Singapore), cells were further mechanically disrupted by passing the lysate through a 26g size needle. Samples were then heated to 95˚C for 5 min. 15 µl of cell lysate proteins were then separated in a 4–12% (w/v) NuPAGE Bis-Tris protein gel and transferred to PVDF membranes. Membranes were incubated with Tris-buffered saline, TBS (50 mM Tris, 150 mM NaCl, pH 7.5) containing 0.1% (v/v) Tween-20 (TBST) and 5% (w/v) BSA for 1 h at room temperature. Membranes were incubated with 1:1000 for mouse α-LAMP1 (ab25630, Abcam, Cambridge, UK), 1:1000 mouse mAb α-LAMP1 (D4O1S, Cell Signaling Technology, USA) 1:1000 for rabbit α-EEA1-Alexa Fluor 647 (ab196186, Abcam, Cambridge, UK), 1:1000 for rabbit α-M6PR-568 (ab202535, Abcam, Cambridge, UK), 1:1000 for rabbit α-cathepsin D (ab75852, Abcam, Cambridge, UK), 1:1000 for rabbit α-Rab5-Alexa Fluor 488 (ab270094, Abcam, Cambridge, UK), 1:1000 for rabbit α-Rab7 (ab137029, Abcam, Cambridge, UK), or 1:1000 for rabbit α-GADPH (5174, Cell Signaling Technology) in TBST containing 1% (w/v) BSA overnight at 4˚C. Membranes were washed for 60 min with TBST at room temperature and then incubated for 2 h at room temperature with goat anti-rabbit (H+L) or goat anti-mouse HRP-linked secondary antibodies (Invitrogen) respectively. After incubation, membranes were washed with TBST for 30 min and specific protein bands were detected by chemiluminescence using SuperSignal West Femto maximum sensitivity substrate (Thermo Fisher Scientific, Singapore). Band intensities were quantified relatively to the lane's loading control using Fiji [82].

**Intracellular reinfection assay.** Infection of keratinocytes was performed in T175 flasks (Nunc; Thermo Fisher Scientific, Singapore) to harvest intracellular bacteria. The infections were performed similarly as described above, except that keratinocytes were infected at MOI 50 for 3 h and subsequently incubated with gentamicin and penicillin G for 1 h. After disruption of keratinocytes, lysates were collected to harvest the intracellular bacteria. Cell lysates were spun down at $100 \times g$ for 1 min to remove debris and the supernatant, which contained the intracellular bacteria, was transferred into a new tube. Harvested bacteria were washed once in PBS and resuspended in complete DMEM. An aliquot of the bacterial suspension was then used for CFU enumeration. The remainder of the bacterial suspension was used for a second round of infection on keratinocyte monolayers in 6-well plates. To achieve a sufficient MOI with recovered intracellular bacteria, each well of a 6-well plate was infected with intracellular-derived bacteria harvested from a T175 flask. For re-infection studies, keratinocytes were similarly infected with bacteria for 3 h and incubated with gentamicin and penicillin G for 1 h. After disruption of keratinocytes, intracellular bacteria were enumerated and the recovery ratio was determined by calculating the ratio between the inoculum CFU to the recovered intracellular CFU. Parallel infections with planktonically grown bacteria as the inoculum were performed by growing bacteria in complete DMEM for 4 h, before washing once in 0.1% Triton X–100 and a second time in PBS. After resuspending the planktonic bacteria in complete DMEM, bacterial cultures were normalized for infection of keratinocytes at MOI 1, 0.1 and 0.01. For the quantification of total cell-associated bacteria, host cells were infected with intracellular-derived bacteria for 3 h and subsequently lysed for CFU enumeration

without prior antibiotic treatment. CFU counts of cell-associated bacteria were normalized against bacterial CFU counts in the supernatant from the same infected wells.

**Statistical analysis.** Statistical analysis was done using Prism 9.2.0 (Graphpad, San Diego, CA). We used one- or two-way analysis of variance (ANOVA) with appropriate post tests, as indicated in the figure legend for each figure, to analyze experimental data comprising 3 independent biological replicates, where each data point is typically the average of 3 technical replicates (unless otherwise noted). In all cases, a p value of $\leq 0.05$ was considered statistically significant.

## Supporting information

**S1 Fig. (related to Fig 2). Intracellular *E. faecalis* is not cell type specific and persists for up to 72 hpi. (A,F)** Enumeration of CFU for OG1RF was performed at different steps of the antibiotic protection assay on HaCaT cells to determine the number of bacteria found intracellularly, compared to the number of bacteria found in the supernatant and final PBS wash after antibiotic treatment. (A) reflects 3 h of infection followed by 1 h of antibiotic treatment. CFU in (F) were enumerated CFU after 3 h of infection and 21 h of antibiotic treatment when optimal killing for strain V583 is achieved. **(B,G)** Enumeration of CFU after antibiotic killing in planktonic cultures of OG1RF and V583 in DMEM + 10% FBS. Planktonic cultures were grown for 3 h at an inoculum size equivalent to MOI 100 from the antibiotic protection assay prior to addition of antibiotics. Cultures were incubated in the presence of antibiotics for either 1 h or 21 h at 37°C with 5% $CO_2$. Bacteria were pelleted and resuspended in sterile 1×PBS to remove residual antibiotics before CFU enumeration. For OG1RF at 1 h and 21 h post antibiotic treatment, and V583 at 21 h post antibiotic treatment, zero CFU counts were observed when bacteria were not resuspended in 1×PBS before CFU enumeration. **(C)** Solid lines indicate the mean CFU at 2–4 hpi at MOI 100 from at least 3 independent experiments. **(D,E)** HaCaTs were infected with *E. faecalis* OG1RF and V583 at MOI 100 for 3 h, followed by treatment with gentamicin and penicillin for 1, 21, 45, 69 h before lysis to obtain the intracellular population. Solid lines indicate the mean CFU from at least 2 independent experiments. Dashed lines serve as point of reference for $10^4$ CFU, for easy visualization of the comparative increase in V583 CFU.
(TIF)

**S2 Fig. (related to Fig 2). Viability of HaCaT cells upon infection with *E. faecalis* OG1RF.** HaCaT cells were infected with MOI 100 of *E. faecalis* OG1RF for 3 h and incubated with 500 μg/ml of gentamicin and penicillin up to 69 hpi and subsequently assessed for viability using the AlamarBlue cell viability reagent.
(TIFF)

**S3 Fig. (related to Fig 2). *E. faecalis* entry into keratinocytes is not dependent on microtubule polymerization, clathrin- and caveolae-mediated endocytosis.** Keratinocytes were pretreated with **(A)** microtubule inhibitor colchicine (10 μg/ml), **(B)** dynasore, an inhibitor of the large GTPase dynamin that is important for the formation of clathrin-coated vesicles (80) (25 μg/ml), or **(C)** nystatin, which selectively affects caveolae-mediated endocytosis by binding sterols, causing caveolae and cholesterol disassembly in the plasma membrane (81, 82) (25 μg/ml). Cells were pre-treated with compounds for 0.5 h and then infected with *E. faecalis* at MOI 100 for 1, 2, or 3 h. For enumeration of intracellular CFU, each infection period was followed by 1 h antibiotic treatment, for a total of 2, 3 or 4 hpi. Adherent or intracellular bacteria were enumerated at the indicated time points (only significant differences are indicated). Solid lines indicate the mean for each data set of at least 3 independent experiments. **p<0.01 2 way

ANOVA, Sidak's multiple comparisons test.
(TIF)

**S4 Fig. (related to Fig 3).** *E. faecalis* **at the periphery of keratinocytes at 24 hpi.** CLSM representative images of infected keratinocytes with condensed nuclei following 3 h of infection and 21 h of incubation in antibiotic laced media. Blue, dsDNA stained with Hoechst 33342; green, E-GFP *E. faecalis*; red, F-actin. Data shown are representative of at least 3 independent experiments.
(TIFF)

**S5 Fig. (related to Fig 3). Non-replicating** *E. faecalis* **do not incorporate BrdU nor RADA.** Fluorescent *E. faecalis* (pDasherGFP) was treated with the antibiotic ramoplanin to halt replication. **(A)** BrdU and **(B)** RADA labelling of bacteria in presence or absence of 26 μg/ml ramoplanin for 1 h. Scale bar: 2 μm.
(TIFF)

**S6 Fig. (related to Fig 3). Replicating and non-replicating intracellular** *E. faecalis* **in** *ex vivo* **cells isolated from infected wounds.** (A) CLSM view of *ex vivo* murine wound tissue cells following infection and BrdU treatment. Left panel shows multiple examples of potentially replicating *E. faecalis* clusters, indicated with white arrows. Scale bar: 10 μm. (B) Enlarged area within white box in (A) on the top, and the same area with the Hoechst channel removed for clear viewing of the other markers on the bottom. The marked areas with white squares show CD45-negative *E. faecalis* containing cells. Scale bar: 2 μm. (C) Enlarged areas within white boxes in B show examples of non-replicating and replicating *E. faecalis*. Blue, dsDNA stained with Hoechst 33342; green, *E. faecalis*; red, BrdU; white, CD45. Images are representative of 3 independent experiments.
(TIFF)

**S7 Fig. (related to Fig 4). Most Rab5 and Rab7 compartments in** *E. faecalis* **infected keratinocytes do not colocalize with** *E. faecalis***-containing compartment.** CLSM Orthogonal views and individual channels of *E. faecalis* within keratinocytes labelled with antibodies against Rab5 (alone, left panels) or together with Rab7 (right panels) at 30 min, 1 h and 3 hpi. Left Panels: White, F-actin; green, *E. faecalis* (pDasherGFP); red, Rab5. Right panels: white, dsDNA stained with Hoechst 33342; green, *E. faecalis* (pDasherGFP); red, Rab7; blue, Rab5. Images are representative of 3 independent experiments. Scale bar: 10 μm.
(TIFF)

**S8 Fig. (related to Fig 4).** *E. faecalis* **is found in heterogeneously labelled Rab7/LAMP1 compartments. (A)** CLSM of infected HaCaTs with fluorescent labelling of Rab7 (late endosome) and fluorescent *E. faecalis* (pDasherGFP). Images show examples of Rab7+ and Rab7- compartments. Green, *E. faecalis* (pDasherGFP); and red, Rab7. Images shown are representative of 3 independent experiments. Scale bar: 5 μm. **(B)** CLSM of infected HaCaTs with fluorescent labelling of Rab7 and LAMP1 (late endosome) and fluorescent *E. faecalis* (pDasherGFP). Pink, *E. faecalis* (pDasherGFP); yellow, Rab7; red, LAMP1. Images show examples of Rab7+/LAMP1- at 4 hpi (top panel), Rab7+/LAMP1+ (middle panel), LAMP1+/Rab7- and LAMP1+/Rab7- (Bottom panel) compartments. Images shown are representative of 3 independent experiments. Scale bar: 5 μm. White arrows indicate areas of interest for *E. faecalis*-containing compartments.
(TIFF)

**S9 Fig. (related to Fig 5).** *E. faecalis* **is rarely found in compartments that contain Cathepsin D. (A)** CLSM Orthogonal views and individual channels of *E. faecalis* within keratinocytes

labelled with antibodies against Cathepsin D and LAMP1 (monoclonal antibody) at 4 h and 24 hpi. Examples of *E. faecalis* colocalizing with LAMP1 but not with Cathepsin D can be observed (white arrows). White, dsDNA stained with Hoechst 33342; green, *E. faecalis* (pDasherGFP); red, LAMP1; and blue, Cathepsin D. Images are representative of 3 independent experiments. Scale bar: 10 μm. **(B)** Rare example of *E. faecalis* within keratinocyte colocalizing with Cathepsin D (white arrow). Keratinocytes were labelled with antibodies against Cathepsin D and LAMP1 (monoclonal antibody) at 24 hpi. White, dsDNA stained with Hoechst 33342; green, *E. faecalis* (pDasherGFP); red, LAMP1; blue, Cathepsin D. Images are representative of 3 independent experiments. Scale bar: 10 μm.
(TIFF)

**S10 Fig. (related to Fig 5). Internalized *E. faecalis* persist within late endosomal compartments.** CLSM of infected HaCaTs stained with antibodies against M6PR (late endosome) and LAMP1 (late endosome/lysosome; polyclonal antibody) at 24 hpi. Images are maximum intensity projections of 4–5 optical sections (~2 μm z-volume) and are representative of 3 independent experiments. Scale bar: 2 μm.
(TIFF)

**S11 Fig. (related to Fig 6). *E. faecalis* infection of keratinocytes does not alter expression of other endosomal proteins. (A)** Whole cell lysates analyzed by immunoblot with antibodies α-M6PR, α-EEA1, α-CathepsinD, and α-GAPDH. HaCaT cells were incubated with (+) and without (-) *E. faecalis* OG1RF for 4 hpi and 24 hpi. Images shown are representative of 3 biological replicates. **(B)** Whole cell lysates analyzed by immunoblot with monoclonal antibody α-LAMP1 and α-GAPDH. HaCaT cells were incubated with (+) and without (-) *E. faecalis* OG1RF and V583 for 4 hpi and 24 hpi. Images shown are representative of 5 biological replicates. **(C)** Relative density of the bands of interest were normalized against loading control (GAPDH). Error bars represent biological replicates and mean ± SEM from at least 3 independent experiments. Statistical analysis was performed using unpaired T-test with Welch's correction.
(TIFF)

**S12 Fig. (related to Fig 7): Correlative light and electron microscopy of *E. faecalis* infected keratinocytes. (A)** Spinning disk confocal microscopy and correlative TEM of HaCaTs stably expressing LAMP1-mCherry infected with *E. faecalis*-GFP at 18 hpi. Confocal images are maximum intensity projections of 4–5 optical sections (~2 μm z-volume). **(B)** Enlarged views of area 1 highlighted in (A). **(C)** Serial section TEM and 3D surface rendering of the area shown in (B). **(D)** Enlarged views of area 2 highlighted in (A). **(E)** Serial section TEM and 3D surface rendering of the area shown in (D). Large arrowheads indicate *E. faecalis* containing vacuoles, small arrows indicate LAMP1+ compartments. VM: vacuolar membrane (VM); MLB: multilamellar body. An *E. faecalis* containing vacuole containing a LAMP1+ve MLB is shown in (B and C), while the *E. faecalis* containing vacuole shown in (D and E) is LAMP1-ve and does not contain an MLB (data pertinent to Fig 5F-H).
(TIF)

**S13 Fig. (related to Fig 7): Ultrastructure of *E. faecalis* containing vacuoles in infected keratinocytes. (A)** Representative high magnification TEM images of *E. faecalis* containing vacuoles. Intact and partially intact bacteria are shown. Two examples of vacuoles containing MLBs are shown. **(B and C)** Serial section TEM analysis of *E. faecalis* containing vacuoles. **(B)** Two *E. faecalis* residing in a shared vacuole (area identical to that shown in Fig 7D). Note the continuity of the vacuolar lumen indicated by the two arrowheads. **(C)** Two *E. faecalis* residing in separate vacuoles. Note that the two vacuoles are separated by a vacuolar membrane,

indicated by the two arrowheads.
(TIF)

**S1 Table. Viability of HaCaT cells upon treatment with inhibitors.** HaCaT cells were incubated with various pharmacological inhibitors at the concentration used in antibiotic protection assays and subsequently assessed for viability using the AlamarBlue cell viability reagent. For cytochalasin D and latrunculin A, cells were incubated with the inhibitor for 24 h prior to assessment of viability. For wortmannin, colchicine, nystatin and dynasore, cells were incubated with the inhibitor for 4 h. Inhibitors resulting in HaCaT viability above 80% were considered as non-cytotoxic.
(DOCX)

**S2 Table. Primers used in this study.**
(DOCX)

**S1 Video. (related to Fig 3). Representative example of BrdU labelling of *ex-vivo* murine wound tissue cells infected with fluorescent *E. faecalis*.**
(MP4)

## Acknowledgments

We thank Kevin B. Wood from the University of Michigan for kindly providing the constitutively expressing GFP plasmid for *E. faecalis* (pDasherGFP) used in this study. We also thank Sng Wan Xin and Heng Yi Ting Eunice for technical assistance in this project. We are grateful to Kline lab members Haris Antypas, Claudia Stocks, Chor Ming Thong, and Brenda Tien for critical feedback on the manuscript.

## Author Contributions

**Conceptualization:** Wei Hong Tay, Kimberly A. Kline.

**Data curation:** Ronni A. G. da Silva, Foo Kiong Ho, Kelvin K. L. Chong, Alexander Ludwig.

**Formal analysis:** Ronni A. G. da Silva, Wei Hong Tay, Foo Kiong Ho, Frederick Reinhart Tanoto, Alexander Ludwig.

**Funding acquisition:** Alexander Ludwig, Kimberly A. Kline.

**Investigation:** Ronni A. G. da Silva, Frederick Reinhart Tanoto, Kelvin K. L. Chong, Alexander Ludwig, Kimberly A. Kline.

**Methodology:** Ronni A. G. da Silva, Wei Hong Tay, Foo Kiong Ho, Frederick Reinhart Tanoto, Kelvin K. L. Chong, Pei Yi Choo, Alexander Ludwig, Kimberly A. Kline.

**Project administration:** Kimberly A. Kline.

**Resources:** Kimberly A. Kline.

**Supervision:** Alexander Ludwig, Kimberly A. Kline.

**Validation:** Ronni A. G. da Silva, Frederick Reinhart Tanoto.

**Visualization:** Ronni A. G. da Silva, Wei Hong Tay, Alexander Ludwig.

**Writing – original draft:** Ronni A. G. da Silva, Wei Hong Tay, Foo Kiong Ho, Kelvin K. L. Chong, Alexander Ludwig, Kimberly A. Kline.

**Writing – review & editing:** Ronni A. G. da Silva, Wei Hong Tay, Foo Kiong Ho, Frederick Reinhart Tanoto, Kelvin K. L. Chong, Pei Yi Choo, Alexander Ludwig, Kimberly A. Kline.

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
