## [Decision Letter · Decision Letter 0]

8 Nov 2021

Dear Kline,

Thank you very much for submitting your manuscript "Enterococcus faecalis persists and replicates within epithelial cells in vitro and in vivo during wound infection" for consideration at PLOS Pathogens. As with all papers reviewed by the journal, your manuscript was reviewed by members of the editorial board and by several independent reviewers. In light of the reviews (below this email), we would like to invite the resubmission of a significantly-revised version that takes into account the reviewers' comments.

Although the reviewers find that the manuscript addresses interesting and clinically relevant questions related to the persistence of Enterococci during infection, they also raise a number of concerns that need to be addressed in a revised manuscript. The following issues would be particularly important to address:

1. The authors use the classical gentamicin exclusion assay to assess intracellular bacteria. As indicated by reviewer 1 and 3, no data is presented to show that extracellular bacteria are killed in your specific model system. Such verification needs to be added to the manuscript to enable proper interpretation and conclusions of the results.

2. As indicated by all three reviewers, the number of replicates used for analyses are generally low and due to their spread in several assays this makes interpretation difficult and some of the conclusion premature.

3. There is insufficient evidence to conclude that Enterococci replicate inside cells. First, the criteria used during imaging to determine whether bacteria are intracellular or not are clear. This combined with the reported decrease in intracellular bacteria over time, a decrease of bacteria in LAMP-positive vacuoles over times, and lack of evidence for replication in the CLEM analyses do not support the conclusion that bacteria replicate intracellularly. All three reviewers suggest approaches to address this issue.

4. In the experiments addressing bacterial entry mechanisms, information showing that the inhibitors used are not toxic to cells or bacteria (except for Dynasore) and that they function as specified in your specific cell system would be needed to draw clear conclusions.

5. Finally, all reviewers request additional information to clarify the results on intracellular trafficking and manipulation of Rab5 and Rab7. Information about how co-localization was determined is missing and how the authors have excluded the possibility that potential transient markers, such as Rab5 and Rab7 were never associated with the bacterial vesicles are not stated. This will be important to specify.

We cannot make any decision about publication until we have seen the revised manuscript and your response to the reviewers' comments. Your revised manuscript is also likely to be sent to reviewers for further evaluation.

Sincerely,

Anders P Hakansson, Ph.D.

Associate Editor

PLOS Pathogens

Michael Wessels

Section Editor

PLOS Pathogens

Kasturi Haldar

Editor-in-Chief

PLOS Pathogens

orcid.org/0000-0001-5065-158X

Michael Malim

Editor-in-Chief

PLOS Pathogens

orcid.org/0000-0002-7699-2064

Your paper has now been reviewed by three experts in the field. Although the reviewers find that the manuscript addresses interesting and clinically relevant questions related to the persistence of Enterococci during infection, they also raise a number of concerns that need to be addressed in a revised manuscript. The following issues would be particularly important to address:

1. The authors use the classical gentamicin exclusion assay to assess intracellular bacteria. As indicated by reviewer 1 and 3, no data is presented to show that extracellular bacteria are killed in your specific model system. Such verification needs to be added to the manuscript to enable proper interpretation and conclusions of the results.

2. As indicated by all three reviewers, the number of replicates used for analyses are generally low and due to their spread in several assays this makes interpretation difficult and some of the conclusion premature.

3. There is insufficient evidence to conclude that Enterococci replicate inside cells. First, the criteria used during imaging to determine whether bacteria are intracellular or not are clear. This combined with the reported decrease in intracellular bacteria over time, a decrease of bacteria in LAMP-positive vacuoles over times, and lack of evidence for replication in the CLEM analyses do not support the conclusion that bacteria replicate intracellularly. All three reviewers suggest approaches to address this issue.

4. In the experiments addressing bacterial entry mechanisms, information showing that the inhibitors used are not toxic to cells or bacteria (except for Dynasore) and that they function as specified in your specific cell system would be needed to draw clear conclusions.

5. Finally, all reviewers request additional information to clarify the results on intracellular trafficking and manipulation of Rab5 and Rab7. Information about how co-localization was determined is missing and how the authors have excluded the possibility that potential transient markers, such as Rab5 and Rab7 were never associated with the bacterial vesicles are not stated. This will be important to specify.

Reviewer's Responses to Questions

**Part I - Summary**

Reviewer #1: This paper from Tay et al. explores the interesting issue of how Enterococcus faecalis persists chronically in wounds, which is important knowledge that could be leveraged for developing novel treatments in what is a very important clinical indication from a healthcare burden perspective. As the authors themselves recognize, the main results are not completely new – a number of other publications have shown that E. faecalis invades and persists in different kinds of cells (both immune and non-immune). The diversity of strategies that might be used by E. faecalis to persist inside keratinocytes is the novelty, but clarification and a number of confirmatory experiments are required to support these findings. As the results stand, the conclusions are not supported. The risk is that if those experiments are improved, it might be the case that the main (novel) conclusions are not supported.

Reviewer #2: In this manuscript, Kline and colleagues detail the intracellular lifestyle of Enterococcus faecalis, a Gram-positive commensal bacterium of the digestive tract. Enterococci are also opportunistic pathogens, colonizing the mouth and wounds. Widely considered as an extracellular pathogen, a number of earlier studies have indicated that E. faecalis can also reside intracellularly. Here the authors use a keratinocyte cell line, HaCaT, as an in vivo model of wound infections. Two strains of E. faecalis, OG1RF and V583, enter these cells, and persist. Bacterial entry is dependent on polymerization of actin cytoskeleton, suggestive of micropinocytosis. Labeling with endocytic markers suggest that maturation of vacuole-residing enterococci follows the endocytic pathway. The correlative and light electron microscopy is a nice addition to the conventional fluorescence microscopy, although the number of bacteria examined is very low. The paper is well-written and easy to follow and the figures are nicely presented. The conclusions are for the most-part justified, with the exception of some concerns that I have detailed below.

Reviewer #3: The authors describe the ability for E. faecalis to invade immune and non-immune (namely keratinocytes) cells during a murine wound infection model. This phenomenon, as with other sites of the body, appears to confer significant biological advantages for the microbe, allowing it to persist in spite of a robust immune response and potentially traditional antimicrobial treatments. The authors performed an extensive and well selected set of experiments in order to begin unravelling the mechanisms of invasion. This manuscript is an interesting, important and well written piece of work. However, there are a number of minor issues that I believe require further explanation and clarification before publication.

**Part II – Major Issues: Key Experiments Required for Acceptance**

Reviewer #1: Most of the experiments needed are presented in the paper, so it's not a case that major experiments are needed, but the issue is that a lot of them are not rigorous enough and would have to be repeated/controls included/more replicates etc. If those experiments are improved, it might be the case that the main (novel) conclusions are not supported.

1. There is insufficient evidence to support the idea that E. faecalis replicates inside cells. CFUs go down over time; many experiments do not have parallel confocal stacks to confirm intracellularity during the experiment; that bacteria may merely be re-entering anew (rather than persisting) has not been ruled out; some experiments suffer from a lack of information about MOI, antibiotic concentration and kill curves, and bacteria growth curves; there are insufficient controls to cover the well-known pitfalls of the protection assay (including incomplete bacterial killing; cell permeability at high concentrations; and protection due to extracellular biofilms). The details for the above observations are provided in Part III.

2. The inhibitor and co-localization data are not convincing - The details for the above observations are provided in Part III.

3. A few experiments do not have enough biological replicates to give confidence that the findings are robust - The details for the above observations are provided in Part III.

Reviewer #2: Points of concern to be addressed:

(1) Figure 2A and 2B. Very few of the total bacterial population are being internalized into keratinocytes (from my estimate comparing Figure 2A with 2B, it is in the order of 1-5%). Is this internalization event an E. faecalis driven event or do keratinocytes have some phagocytic-like activity that allow them to internalize bacteria? Comparing the internalization of E. faecalis with non-pathogenic E. coli (DH10B or K12, for example), or an "extracellular" Gram-positive bacterium, would answer this question.

(2) Figure 2C and 2D. Compared to total CFUs (Supp Fig 1), which show a steady-state level of viable intracellular CFUs from 4-48 h, microscopy images show that there is an increase in the number of bacteria/cell with time. This increase cannot be in all infected cells, otherwise there would be an increase in total CFUs. In what percentage of infected cells does this increase in bacterial numbers occur? It is important to know what proportion of keratinocytes support bacteria replication. The authors should count bacteria in individual cells (i.e. single-cell analysis) over a time course to chart the heterogeneity in the intracellular distribution of bacteria. I suspect there are multiple scenarios happening, with the total CFUs being the sum of (i) bacterial replication in some cells, (ii) no replication in some cells, (iii) bacterial death in some cells and (iv) keratinocyte cell death.

(3) Related to trafficking data and the model depicted in Figure 8. The way that the authors do the infection, it is very asynchronous i.e. with the extended 3 h time period that the bacteria are in contact with the host cells (prior to antibiotic addition) so bacteria will enter almost immediately and others 3 h later. This creates a huge time spread in the trafficking of intracellular bacteria, which complicates the analysis of host cell marker acquisition. I understand that a long infection time is required to increase the number of intracellular bacteria, so I am not faulting the experimental design, but this caveat should be acknowledged. It seems odd to assess EEA1 acquisition at 4 h post-infection when EEA1, an early endosome marker, would only be acquired 15-30 min post-entry. For this reason, it is not surprising that so few bacteria are labelled with this early endosome marker after 4 h. Rab7 is transiently acquired by late endosomes and is lost upon their transition into lysosomes. Its transient nature of association makes it difficult to assess whether a vesicle/phagosome/bacterium has ever acquired Rab7 or not, unless live-cell imaging is undertaken. If a bacterium is negative for Rab7, Rab7 may have been acquired and lost, for example. As depicted in Figure 8, can the authors show that bacterial viability is associated with Rab7/LAMP1-labelling (scenario I, II or III in Figure 8)? Bacteria containing an inducible fluorescent protein could easily be used to assess viability in each of the described vacuoles.

(4) The CLEM data requires number of events scored to validate statements such as “all membrane-bound E. faecalis were spatially separated from lysosomes, and there was no indication of membrane fusion between E. faecalis-containing compartments and lysosomes”. Specifically, “we observed E. faecalis in LAMP1 positive compartments as well as in vacuoles that appeared to be devoid of LAMP1” please provide percentages; “most internalized bacteria appeared to be morphologically intact”, what proportion?; “we did not find evidence of multiple replicating E. faecalis within a single LAMP1 positive compartment leading to membrane distension” how many bacteria were assessed?; “In some instances, however, vacuoles harbouring E. faecalis appeared to contain LAMP1 positive multi-lamellar bodies (MLBs)”, how many instances?

(5) Figure 5B. LAMP1 is a heavily glycosylated protein that migrates at ~100 kDa on SDS-PAGE gels. The image shown in Figure 5B suggests that the LAMP1 antibody used in this study is not actually specific for LAMP1. Monoclonal H4A3 (available from Developmental Studies Hybridoma Bank) and D4O1S (Cell Signaling, https://www.cellsignal.com/products/primary-antibodies/lamp1-d4o1s-mouse-mab/15665) are validated antibodies for LAMP1. I would like to see that the LAMP1 data is reproducible for either of these antibodies. Given that the authors are drawing conclusions about the steady-state levels of endocytic proteins, it is imperative that they are actually detecting LAMP1 here.

Reviewer #3: If possible the authors should present the gentamicin protection assay final wash CFU's. The addition of this data would significantly improve this manuscript

**Part III – Minor Issues: Editorial and Data Presentation Modifications**

Reviewer #1: (I have pulled out and mentioned key issues in the preceding section but I include all of my comments here for ease of reference)

Minor general Comment:

- Statements saying that they are raising the issue that intracellular persistence may influence chronic E. faecalis infection should be removed or modified, since other studies had already raised that hypothesis. Therefore these authors are supporting those works but not trail-blazing the hypothesis.

- The absence of numbered lines in the document was very frustrating and hindered the review process.

Title:

Needs to be altered or qualified, otherwise it is misleading, because:

- The authors don’t have enough evidence to show that E. faecalis replicate in vivo (see more detailed comments below)

- Moreover, they only show one (very limited) experiment “during wound infection” in mice. If they wanted to focus the paper on wound infection, additional experiments are necessary to confirm the results (e.g. with epithelium organoids, or more mice experiments, or mimicking wounds in vitro in cell layers). In opposition, the majority of the results were obtained with cells/monolayers of keratinocytes without mimicking a wound.

Introduction:

This section is extremely short and superficial, focusing mainly on the discussion of a previous paper published by the authors and a summary of the results obtained in this work. It would have been good to see more details of other papers showing Enterococcus invasion to set the scene. Also, it is always good to give the clinical picture – why should we care about wound infection? What is the global burden both economically and from an incidence point of view?

Minor comment: Enterococci should not be in italics.

Results:

Intracellular E. faecalis are present within CD45+ and CD45- cells during mouse wound infection

- The antibiotic protection assay is not without its flaws. First, the antibiotic concentration used should be mentioned (at least as Supplementary material) and a dose response curve of antibiotic(s) used should be shown to assess if the concentration used was enough to kill all the bacteria. On the other hand, previous work by others has shown that very high concentrations of so-called impermeant antibiotics (even the gold standard gentamicin) can actually enter some cell types – have the authors confirmed that this does not happen under their conditions? Does the antibiotic have any cytotoxic effect on the suspended cells? We need to know this to assess the results obtained regarding the intracellular bacteria quantification. Finally, and perhaps most critically, Enterococci can form biofilms, even in a relatively short period of time, and biofilms would be largely resistant to antibiotics. Extracellular biofilms therefore could contribute to CFU post-treatment which could artificially inflate the estimates of intracellular bacteria.

- The area/depth of the wound harvested and the mean number of cells recovered should also be mentioned (in the Material & Methods and/or main text), to help the reader understand possible variations in the CFU counting.

- A comment/discussion or possible explanation observed for the 2 different “subpopulations” of infected CD45- cells should be provided. This becomes important in light of the subsequent results obtained in vitro by the authors.

E. faecalis adheres to and enters keratinocytes

- As above, a dose response curve of antibiotic(s) used should be shown to assess if the concentration and treatment periods used (which varied from 1h to 21h depending on the experiment) were enough to kill all the bacteria

- The sentence “Parallel cytotoxicity …. (data not shown)” is vague and could be confusing. It should be better explained, since the subsequent experiments in Supp Fig. 1 and Fig. 2 were performed at 1, 2, 3 and 4 hpi.

- The authors’ use of the strain V583 should be better contextualized and explained. It only appeared in this section (and Fig. 5) and the authors commented that it showed “even higher numbers within HaCat cells”, although the bacteria recovered after 48h and 72h dropped dramatically (which is not explained). In addition, since it seems a better “persister”, why was not used for all the other studies?

- The MOIs used for the images presented in Fig 2 should be mentioned (main text and caption).

- A panel of images comparing the different MOIs and time points showed in Fig 2A and 2B should be presented, as well as showing in Z-stack what the authors consider to be “internalized” and “adhered” (e.g. with arrows). Moreover, given there’s some point-spread going on, not all of the bacterial in the ortho views seem, in my opinion, to look particularly internalized at this resolution. As a minor point, it’s quite hard to see the cross-hairs on some images. Maybe make them white?

- In the context of the main results presented in this section, Fig 2C and 2D should be removed or moved to Supplementary, since they show results for much later time points (4 hpi and 24 hpi, respectively).

- The final conclusion that “E. faecalis is replicating within the cells” based on the results obtained after 24 hpi per se seems problematic and should be altered, for two reasons:

o Since the experiments were made in fixed different time points, more bacteria might simply have entered in the meantime and persisted inside the keratinocytes.

o In fig Supp 1 the authors show that intracellular bacterial CFU/keratinocyte does not vary significantly from 4 to 24, 48 and 72 hpi with antibiotic treatment (it even decreases in the latter time point).

- It is difficult to assess the significance of the Supp Fig 2, which was provided to show bacteria on the periphery of the keratinocytes that were previously inside the cells (3 hpi + 21 h antibiotic treatment), because these images also do not show any bacteria inside the cells in Z-stack (+ the bacterial amount is very low from what should be expected according to the authors’ results). Perhaps invasion did not occur at all in this case? And/or the bacteria outside are due to biofilms that resisted this antibiotic concentration (see above)?

Entry of E. faecalis into keratinocytes is dependent on actin polymerization and PI3K

signalling

- Cytotoxicity assessment of all the compounds tested using dose-response curves and/or imaging should be done (or mentioned, if published previously) to validate the results obtained with keratinocytes, since the authors also detected that at least one (Dynasor) that showed cell toxicity. It is probably important to see if they also affect isolated bacteria (or refer to the literature if they’ve been shown to have no effect on prokaryotes).

- Colchicine seems to boost bacterial adhesion and persistence. Any explanation for this?

- Are all the mean differences observed In Supp Fig 3 not statistically significantly different (apart from one time with the Dynasor treatment)? If not, the authors should show the statistical analysis for the others as well.

In case they are non-significant, a comment regarding the high heterogeneity of the results obtained should be made (scale is logarithmic).

- The authors cannot completely exclude a role of the endocytocytis (mediated be claveolae and clathrin), based only in the results from Supp Fig. 3, mainly because:

o They have means based on 3 experiments, which in some cases show high heterogeneity between them (so it suggests that in some cases endocytosis via this pathways might occur).

o Authors comment in Supp Fig. 3 that they saw a large amount of Dynasor-treated cells being killed after 4 hpi (justifying that the reduction in intracellular bacteria is due to keratinocytes’ death). However, they also saw a statistically significant reduction in intracellular bacteria recovered after 3 hpi (and even with 2 hpi you can see the reduction). This supports the possibility that at least in some cases the bacteria are entering via receptor-mediated endocytosis.

o Adhesion of bacteria seems to be impaired with Nystatin (at least in some cases).

o Controls regarding the abolition of the clathrin and claveolae pathways with the inhibitor concentrations tested were not shown or referred by the authors for this particular cells. As this could vary from cell type to cell type, it’s important to confirm that the inhibitors are working as expected.

o The use of Wortmannin per se does not tell us that only macropinocytosis is being influenced; it could also influence phagocytosis, which keratinocytes are also able to do, and even receptor-mediated endocytosis to some extent.

They should either increase their N or try to assess clathrin- and claveolae mediated endocytosis using imaging analysis. Or otherwise consider that receptor-mediated endocytosis pathways might also have a role (at least in some cases).

- Caption from Supp Fig 3 should be reduced and the “discussion” part should be moved to the main text.

Intracellular E. faecalis traffics through early and late endosomes

- No Z-stacks are shown in this section, so how can the authors know for sure that all the bacteria counted was inside? Were these experiments performed in the presence of antibiotics? (If so, dose-response curve should be presented to show the efficacy of the killing, as per above comments)

- Even with prolonged antibiotic treatment, the authors showed previously that E. faecalis is able to escape keratinocytes (Supp Fig 2). How they can confirm that they are not colocalizing bacteria that are outside or between the cells?

- Authors should also clarify if the total N of bacteria/cell compartments counted per condition is supported by a single image (which it seems to be) or different images. More biological replicates seem to be needed.

- Since the total N of bacteria/cell compartments counted vary considerably according to the different labelings, it becomes difficult to compare the percentages indicated by the author. More replicates are needed.

- According to the data, the percentage of E. faecalis in LAMP1-positive cells after 24 hpi is lower than the one observed in 4 hpi. Therefore, the authors cannot conclude that the bacteria are replicating inside the endosomes as they state in their final conclusion statement. Shouldn’t this value be higher?

- There is not enough evidence about replication in endosomes based on these data. Live cell imaging and/or more cell counting (statistically significant/biological replicates) should be provided.

- The panel in Supp Fig 4 is redundant, since the figures in the upper panel are only in a very slightly different magnification compared with the lower panel (higher difference in magnification or other regions/images should be provided instead).

E. faecalis intracellular infection interferes with Rab5 and Rab7 protein levels

- A large initial part of the paragraph (when comparing to other bacteria) should be moved to the Discussion section.

- Similar comments from the section above, regarding the absence of Z-stacks and small N used, apply to this section as well.

- Some comparisons from the strains V583 and OG1RF should be better contextualized. Are both of them equally fit in the conditions tested? (Growth curves should be provided) Were they always inoculated at the same MOI? (the MOIs in this section should be mentioned)

- Since E. faecalis does not change cathepsin D expression (and it seems that it is not naturally reduced over time) and the Rab7 levels seem to be restored after 24 hpi (as the authors also mention), why there is no fusion with lysosomes? This should be better explained or assessed (e.g. targeting/labeling lysosomes) by the authors.

E. faecalis containing vacuoles do not fuse with lysosomes

- The authors mention that they did not find evidence of replication, which is quite contradictory regarding the previous results obtained. Could this be related with different MOIs and/or time of infection? In any case, this should be better explained.

- Some quantification (with more images/biological samples) is needed and/or experiments targeting lysosomes, in order to conclude the absence of fusion with lysosomes.

- In Fig 6 the authors highlight a compartment with “bacteria with altered appearance”. This should be better explained in the text. The only lysosome showed in the figure is also not in a commonly observed shape for the compartments.

- Fig 6 and Supp Fig 7 seem to be based in the same (very limited) amount of images.

Intracellular E. faecalis is primed for more efficient reinfection

- The authors themselves said that the MOI used for reinfection was very low. This seems quite contradictory, since they also argue that E. faecalis replicates inside the cells over time. Therefore it would be meaningful if a first infection with higher MOIs and/or prolonged time of infection (similar to the ones previously used) was used, in order to recover more bacteria for the reinfection and to correlate better with the rest of the results.

Discussion:

All of the discussion and the E. faecalis persistence model in keratinocytes proposed should be reviewed in light of the aforementioned comments – especially the wound injury focus and in vivo conclusions, the replication of bacteria and the exclusion of receptor-mediated endocytosis in the model. The authors have to somehow account for the fact that there is a reduction over time of the intracellular bacteria recovered in their wound infections in vivo (particularly in CD45- cells), which opposes the idea of replication and persistence of bacteria.

A comment/hypothesis/discussion about how cathepsin D is absent in only 2 out of the 3 possible pathways that they present (and not in all for example) should be made, particularly considering that no change in its cell expression was observed.

Reviewer #2: Minor points:

(1) The source/description of E. faecalis V583 is not listed in Materials and Methods.

(2) Figure 5C. Text size is too small to be easily seen.

(4) Supplementary Figure 1B and C. What is the dashed line?

(5) Fluorescence images. As suggested by a number of journal commentaries (https://pubmed.ncbi.nlm.nih.gov/22379119/;
https://www.ascb.org/science-news/how-to-make-scientific-figures-accessible-to-readers-with-color-blindness/), fluorescence images should be shown in greyscale, where possible, to improve visibility to the human eye and accessibility to those with color blindness. This impacts Figure 4, Supp. Fig 4, Figure 5, Supp. Fig 5.

Reviewer #3: Author Summary

“whereupon it manipulates the endosomal pathway and

expression of trafficking molecules required for endo-lysosomal fusion, enabling E. faecalis

to avoid lysosomal degradation and consequent death.”

Qualify language – data does not necessarily prove that the bacteria manipulates the endosomal pathway

“extracellular pathogen”

Is it still thought of as an extracellular pathogen? Please consider rephrasing

Results

Intracellular E. faecalis are present within CD45+ and CD45- cells during mouse wound

Infection

Were the gentamicin treated “extracellular” bacteria definitely dead (live/dead stain)? Was this strain gent susceptible? In our experience, this assay can be quite misleading – did you plate the final PBS washing steps on agar to ensure no/low growth?

Fig 1. Are you plotting biological or technical replicates? You mention that at least 3 replicates were used. Would it be possible to clarify this in the legend?

E. faecalis adheres to and enters keratinocytes

Again, the results from the gentamicin protection assay can be very misleading. If you have no cfu data from the PBS washes, then perhaps qualify the language.

Nonetheless, the imaging data is compelling and beautifully presented. Perhaps make the orthogonal lines a little clearer for the reader.

Supplementary Fig 2. Convincing images but please increase magnification

Entry of E. faecalis into keratinocytes is dependent on actin polymerization and PI3K

Signalling

Fig. 3 and Supplementary fig. 3. This data is convincing; however, the statistical significance is less so. Are all data points technical or biological repeats? The use of parametric tests for such small datasets is unusual. Was the data normally distributed? If not consider non-parametric tests. Indeed, it might be better to not do tests - the difference can be seen in the graph…..

Intracellular E. faecalis traffics through early and late endosomes

How was this percentage data calculated? Was this done using image analysis software or done manually? This needs some explanation. If analysed manually, then how many cells / experiments were included. The images (particularly the red channel) look to have been increased in brightness.

E. faecalis intracellular infection interferes with Rab5 and Rab7 protein levels

“For instance, Mycobacterium tuberculosis affects

Rab7 recruitment and, consequently, phagosome maturation, by interfering with Rab5

effectors (Saikolappan et al. 2012; Puri, Reddy, and Tyagi 2013). Listeria monocytogenes also

affects Rab7 recruitment by inhibiting Rab5 GDP exchange activity in host cells (Prada-

Delgado et al. 2005). Additionally, Coxiella burnetii can localize to Rab5 and LAMP1 positive

compartments that lacks Rab7 (Ghigo et al. 2009; Ghigo, Colombo, and Heinzen 2012).”

I wonder whether this above section should be moved to the introduction / discussion. Or perhaps shortened. It feels out of place.

As mentioned earlier, it is not clear how colocalization with intracellular compartments was measured. Percentages are stated but the method is not clear. Please add. Without this information it is quite difficult to judge how reliable this data is.

Is the WB data normally distributed? T-test is parametric

E. faecalis containing vacuoles do not fuse with lysosomes

Fig. 6 and supplementary Fig 7. CLEM is very nicely presented and looks convincing. It might be easier for the reader to mention the colours in the legend. It is currently quite difficult to understand

Intracellular E. faecalis is primed for more efficient reinfection

“These results are similar to observations made in S. pyogenes, where longer periods of internalization in macrophages increased recovered CFU during subsequent reinfections (Hertzen et al. 2012).”

The above sections belongs in the discussion

Fig 7. (A) I agree with the use of non-parametric tests for these experiments. However, why did you not use this for the previous tests? The N appears to be comparable….

Discussion

“This is well described for uropathogenic E. coli”

This statement is true. However, this is in murine models of infection. Very little data in humans. Please alter language.

I think it is important to clarify that this a mouse / cell model of infection. Experiments using infected wounds in humans would need to be performed to corroborate this data.

PLOS authors have the option to publish the peer review history of their article (what does this mean?). If published, this will include your full peer review and any attached files.

Reviewer #1: No

Reviewer #2: No

Reviewer #3: No
---

## [Editor Report · Decision Letter 1]

27 Feb 2022

Dear Kline,

Thank you very much for submitting your revised manuscript "Enterococcus faecalis alters endo-lysosomal trafficking to replicate and persist within mammalian cells" for consideration at PLOS Pathogens. As with all papers reviewed by the journal, your manuscript was reviewed by members of the editorial board. Based on the editorial assessment, we are likely to accept this manuscript for publication, providing that you modify the manuscript according to the comments below.

After assessing your responses to the Reviewers’ comments and the accompanying revisions and additions incorporated into the revised manuscript, we find that the revised manuscript represents a major improvement that in all major respects addresses the reviewers’ concerns satisfactorily. There are, however, some minor issues remaining, primarily associated with the novel information included, that could be easily amended, and that would not require any additional experiments. These issues are listed below. As long as these minor revisions and amendments to the current manuscript are done, we would expect to return a favorable decision regarding your manuscript very quickly.

1. Based on the request from the reviewers, additional imaging data has been included to better represent the data and their associated conclusions (Fig 4A-D, 5B-C, S7, S8, S9). However, Figures S7 and S9 have no arrows in the individual panels making it difficult to understand what should be observed. Please add. New Figures 4A-D and 5B-C are relevant additions, yet it is hard to see the staining and co-positioning/co-localization with variable staining intensity of blue, red and green colors and no separate panels for each channel. The histogram analysis it a welcome addition, although without the ability for the reader to visually evaluate co-localization or co-positioning of bacteria and markers, the specific criteria used to score this, as described in the Methods section (lines 983-988) becomes unclear and difficult to follow. For example, in Figure 5B-C, the 4 h time-point, why is the first bacteria in the histogram not considered co-localized with LAMP-1 and in the 24 h section, again why is the first bacterium not considered co-localized with LAMP-1? Is this based on signal intensity or visual information that is not apparent unless each marker is observed separately? Additional clarification of the co-localization/co-positioning analyses in these figures would be necessary.

2. A new Figure 3C-D has been added to directly address intracellular replication by showing BrdU and RADA staining of bacteria inside HaCaT cells as well as in RAW macrophages. For both cell types in Figure 3C, arrows would be needed also in the separate panels to orient the reader to the colocalization events. The RADA staining for the HaCaT cells shows unspecific and dotty staining where no bacteria are found. Please comment or explain.

3. Reviewer 1 and 2 indicated a lack of information about endosomal trafficking inhibitor function. Inhibitor function is indirectly addressed in the response to Reviewer 1 through a list of references of papers that have used the same concentrations in the same cell types by other investigators. This information is, however, not included in the manuscript and the use of concentrations and inhibitors listed on lines 913-915 in the Methods section is not complete. To justify the inhibitor concentrations used in the manuscript, in the absence of experimental validation, please include the specific concentrations used for each inhibitor in the Methods section together with the references used to justify these concentrations.

4. Reviewer 2 highlights that the LAMP-1 antibody used may not be optimal and suggests validated antibodies to be used for confirmation. One of the validated antibodies suggested is now used in new figures Fig 5, S9, S11. However, although this is commented on in the response to the Reviewer, it is not mentioned specifically in the manuscript when different antibodies were used. As validation using these antibodies was an issue, it would be relevant to include antibody information for the LAMP-1 staining either in the relevant figure legends or mention in the Methods section what LAMP-1 antibody was used for what Figure, so that the readers can properly interpret the data.

Sincerely,

Anders P Hakansson, Ph.D.

Associate Editor

PLOS Pathogens

Michael Wessels

Section Editor

PLOS Pathogens

Kasturi Haldar

Editor-in-Chief

PLOS Pathogens

orcid.org/0000-0001-5065-158X

Michael Malim

Editor-in-Chief

PLOS Pathogens

orcid.org/0000-0002-7699-2064

After assessing your responses to the Reviewers’ comments and the accompanying revisions and additions incorporated into the revised manuscript, we find that the revised manuscript represents a major improvement that in all major respects addresses the reviewers’ concerns satisfactorily. There are, however, some minor issues remaining, primarily associated with the novel information included, that could be easily amended, and that would not require any additional experiments. These issues are listed below. As long as these minor revisions and amendments to the current manuscript are done, we would expect to return a favorable decision regarding your manuscript very quickly.

1. Based on the request from the reviewers, additional imaging data has been included to better represent the data and their associated conclusions (Fig 4A-D, 5B-C, S7, S8, S9). However, Figures S7 and S9 have no arrows in the individual panels making it difficult to understand what should be observed. Please, add. New Figures 4A-D and 5B-C are relevant additions, yet it is hard to see the staining and co-positioning/co-localization with variable staining intensity of blue, red and green colors and no separate panels for each channel. The histogram analysis it a welcome addition, although without the ability for the reader to visually evaluate co-localization or co-positioning of bacteria and markers, the specific criteria used to score this, as described in the Methods section (lines 983-988) becomes unclear and difficult to follow. For example, in Figure 5B-C, the 4 h time-point, why is the first bacteria in the histogram not considered co-localized with LAMP-1 and in the 24 h section, again why is the first bacterium not considered co-localized with LAMP-1? Is this based on signal intensity or visual information that is not apparent unless each marker is observed separately? Additional clarification of the co-localization/co-positioning analyses in these figures would be necessary.

2. A new Figure 3C-D has been added to directly address intracellular replication by showing BrdU and RADA staining of bacteria inside HaCaT cells as well as in RAW macrophages. For both cell types in Figure 3C, arrows would be needed also in the separate panels to orient the reader to the colocalization events. The RADA staining for the HaCaT cells shows unspecific and dotty staining where no bacteria are found. Please comment or explain.

3. Reviewer 1 and 2 indicated a lack of information about endosomal trafficking inhibitor function. Inhibitor function is indirectly addressed in the response to Reviewer 1 through an extensive list of references of papers that have used the same concentrations in the same cell types by other investigators. This information is, however, not included in the manuscript and the use of concentrations and inhibitors listed on lines 913-915 in the Methods section is not complete. To justify the inhibitor concentrations used in the manuscript, in the absence of experimental validation, please include the specific concentrations used for each inhibitor in the Methods section together with the references used to justify these concentrations.

4. Reviewer 2 highlights that the LAMP-1 antibody used may not be optimal and suggests validated antibodies to be used for confirmation. One of the validated antibodies suggested is now used in new figures Fig 5, S9, S11. However, although this is commented on in the response to the Reviewer, it is not mentioned specifically in the manuscript when different antibodies were used. As validation using these antibodies was an issue, it would be relevant to include antibody information for the LAMP-1 staining either in the relevant figure legends or mention in the Methods section what LAMP-1 antibody was used for what Figure, so that the readers can properly interpret the data.

Reviewer Comments (if any, and for reference):

Figure Files:

Data Requirements:

Reproducibility:

References:

---

## [Editor Report · Decision Letter 2]

10 Mar 2022

Dear Kline,

We are pleased to inform you that your manuscript 'Enterococcus faecalis alters endo-lysosomal trafficking to replicate and persist within mammalian cells' has been provisionally accepted for publication in PLOS Pathogens.

Best regards,

Anders P Hakansson, Ph.D.

Associate Editor

PLOS Pathogens

Michael Wessels

Section Editor

PLOS Pathogens

Kasturi Haldar

Editor-in-Chief

PLOS Pathogens

orcid.org/0000-0001-5065-158X

Michael Malim

Editor-in-Chief

PLOS Pathogens

orcid.org/0000-0002-7699-2064

---

## [Editor Report · Acceptance letter]

1 Apr 2022

Dear Kline,

We are delighted to inform you that your manuscript, "*Enterococcus faecalis* alters endo-lysosomal trafficking to replicate and persist within mammalian cells," has been formally accepted for publication in PLOS Pathogens.

Best regards,

Kasturi Haldar

Editor-in-Chief

PLOS Pathogens

orcid.org/0000-0001-5065-158X

Michael Malim

Editor-in-Chief

PLOS Pathogens

orcid.org/0000-0002-7699-2064